# Systems genetics in diversity outbred mice inform BMD GWAS and identify determinants of bone strength

Basel M. Al-Barghouthi [1,2,8], Larry D. Mesner[1,3,8], Gina M. Calabrese[1,8], Daniel Brooks[4], Steven M. Tommasini [5], Mary L. Bouxsein [4], Mark C. Horowitz[5], Clifford J. Rosen [6], Kevin Nguyen[1], Samuel Haddox[2], Emily A. Farber[1], Suna Onengut-Gumuscu [1,3], Daniel Pomp[7] & Charles R. Farber [1,2,3 ✉]

Genome-wide association studies (GWASs) for osteoporotic traits have identified over 1000 associations; however, their impact has been limited by the difficulties of causal gene identification and a strict focus on bone mineral density (BMD). Here, we use Diversity Outbred (DO) mice to directly address these limitations by performing a systems genetics analysis of 55 complex skeletal phenotypes. We apply a network approach to cortical bone RNA-seq data to discover 66 genes likely to be causal for human BMD GWAS associations, including the genes SERTAD4 and GLT8D2. We also perform GWAS in the DO for a wide-range of bone traits and identify Qsox1 as a gene influencing cortical bone accrual and bone strength. In this work, we advance our understanding of the genetics of osteoporosis and highlight the ability of the mouse to inform human genetics.

[1] Center for Public Health Genomics, School of Medicine, University of Virginia, Charlottesville, VA, USA. [2] Department of Biochemistry and Molecular Genetics, School of Medicine, University of Virginia, Charlottesville, VA, USA. [3] Department of Public Health Sciences, School of Medicine, University of Virginia, Charlottesville, VA, USA. [4] Center for Advanced Orthopedic Studies, Beth Israel Deaconess Medical Center, Department of Orthopedic Surgery, Harvard Medical School, Boston, MA, USA. [5] Department of Orthopaedics and Rehabilitation, Yale School of Medicine, New Haven, CT, USA. [6] Maine Medical Center Research Institute, Scarborough, ME, USA. [7] Department of Genetics, University of North Carolina School of Medicine, Chapel Hill, NC, USA. [8] These authors contributed equally: Basel M. Al-Barghouthi, Larry D. Mesner, Gina M. Calabrese. ✉email: crf2s@virginia.edu

Osteoporosis is a condition of low bone strength and an increased risk of fracture[1]. It is also one of the most prevalent diseases in the U.S., affecting over 10 million individuals[2]. Over the last decade, efforts to dissect the genetic basis of osteoporosis using genome-wide association studies (GWASs) of bone mineral density (BMD) have been tremendously successful, identifying over 1000 independent associations[3–5]. These data have the potential to revolutionize our understanding of bone biology and the discovery of novel therapeutic targets[6,7]; however, progress to date has been limited.

One of the main limitations of human BMD GWAS is the difficulty in identifying causal genes. This is largely due to the fact that most associations implicate non-coding variation presumably influencing BMD by altering gene regulation[5]. For other diseases, the use of molecular "-omics" data (e.g., transcriptomic, epigenomic, etc.) in conjunction with systems genetics approaches (e.g., identification of expression quantitative trait loci (eQTL) and network-based approaches) has successfully informed gene discovery[8,9]. However, few "-omics" datasets exist on bone or bone cells in large human cohorts (e.g., bone or bone cells were not part of the Genotype-Tissue Expression (GTEx) project[10]), limiting the use of systems genetics approaches to inform BMD GWAS[11].

A second limitation is that all large-scale GWASs have focused exclusively on BMD[3–5]. BMD is a clinically relevant predictor of osteoporotic fracture; however, it explains only part of the variance in bone strength[12–15]. Imaging modalities and bone biopsies can be used to collect data on other bone traits such as trabecular microarchitecture and bone formation rates; however, it will be difficult to apply these techniques at scale ($N \geq 100$ K). Additionally, many aspects of bone, including biomechanical properties, cannot be measured in vivo. These limitations have hampered the dissection of the genetics of osteoporosis and highlight the need for resources and approaches that address the challenges faced by human studies.

The diversity outbred (DO) is a highly engineered mouse population derived from eight genetically diverse inbred founders (A/J, C57BL/6J, 129S1/SvImJ, NOD/ShiLtJ, NZO/HILtJ, CAST/EiJ, PWK/PhJ, and WSB/EiJ)[16]. The DO has been randomly mated for over 30 generations and, as a result, it enables high-resolution genetic mapping and relatively efficient identification of causal genes[17,18]. As an outbred stock, the DO also more closely approximates the highly heterozygous genomes of a human population. These attributes, coupled with the ability to perform detailed and in-depth characterization of bone traits and generate molecular data on bone, position the DO as a platform to assist in addressing the limitations of human studies described above.

In this work, we present a resource for the systems genetics of bone strength consisting of information on 55 bone traits from over 600 DO mice, and RNA-seq data from marrow-depleted cortical bone in 192 DO mice. We demonstrate the utility of this resource in two ways. First, we apply a network approach to the bone transcriptomics data in the DO and identify 66 genes that are bone-associated nodes in Bayesian networks, and their human homologs are located in BMD GWAS loci and regulated by colocalizing eQTL in human tissues. Of the 66, 19 are not previously known to influence bone. The further investigation of two of the 19 novel genes, SERTAD4 and GLT8D2, reveals that they are likely causal and influence BMD via a role in osteoblasts. Second, we perform GWASs in the DO for 55 complex traits associated with bone strength identifying 28 QTL. By integrating QTL and bone eQTL data in the DO, we identify Qsox1 as the gene responsible for a QTL on Chromosome (Chr.) 1 influencing cortical bone accrual along the medial–lateral femoral axis and femoral strength. These data highlight the power of the DO

mouse resource to complement and inform human genetic studies of osteoporosis.

## Results

**Development of a resource for the systems genetics of bone strength.** An overview of the resource is presented in Fig. 1. We measured 55 complex skeletal phenotypes in a cohort of DO mice ($N = 619$; 314 males, 305 females; breeding generations 23–33) at 12 weeks of age. We also generated RNA-seq data from marrow-depleted femoral diaphyseal bone from a randomly chosen subset of the 619 phenotyped mice ($N = 192$; 96/sex). All 619 mice were genotyped using the GigaMUGA[19] array (~110 K SNPs) and these data were used to reconstruct the genome-wide haplotype structures of each mouse. As expected, the genomes of DO mice consisted of approximately 12.5% from each of the eight DO founders (Fig. 2a).

The collection of phenotypes included measures of bone morphology, microarchitecture, and biomechanics of the femur, along with tibial histomorphometry and marrow adiposity (Supplementary Data 1 and 2). Our data included quantification of femoral strength as well as many clinically relevant predictors of strength and fracture risk (e.g., trabecular and cortical microarchitecture). Traits in all categories (except tibial marrow adipose tissue (MAT)) were significantly ($P_{adj} < 0.05$) correlated with femoral strength (Supplementary Data 3). Additionally, all traits exhibited substantial variation across the DO cohort. For example, we observed a 30.8-fold variation (the highest measurement was 30.8 times greater than the lowest measurement) in trabecular bone volume fraction (BV/TV) of the distal femur and 5.6-fold variation in femoral strength (Fig. 2b). After adjusting for covariates (age, DO generation, sex, and body weight) all traits had non-zero heritabilities ($h^2$) (Fig. 2c). Correlations between traits in the DO were consistent with expected relationships observed in previous mouse and human studies (Supplementary Data 4)[20–23].

In addition to standard RNA-seq quality control procedures (see "Methods" section), we also assessed RNA-seq quality by principal components analysis (PCA) and did not observe any major effect of sex, batch, and age in the first two principal components, which explained over 50% of the variance (Supplemental Fig. 1). We did observe a separation of samples based on sex in the third PC, but it only explained 2.4% of the variance. Importantly, our PCA analysis did not identify any outliers in the bulk RNA-seq data. Furthermore, we performed differential expression analyses between sexes and between individuals with high versus low bone strength (Supplementary Data 5 and 6). As expected, the most significantly differentially expressed genes based on sex were located on the X chromosome. We identified 83 significantly (FDR < 0.05) differentially-expressed transcripts in the analysis of low and high bone strength. Many were genes, such as Ahsg[24] and Arg1[25], which have previously been implicated in the regulation of bone traits.

**Identification of bone-associated nodes.** We wanted to address the challenge of identifying causal genes from BMD GWAS data, using the DO resource described above. To do so, we employed a network-based approach similar to one we have used in prior studies[26,27] (Fig. 3). First, we partitioned genes into groups based on co-expression by applying weighted gene co-expression network analysis (WGCNA) to the DO cortical bone RNA-seq data[28]. We generated three WGCNA networks; sex-combined, male, and female. The three networks contained a total of 124 modules (Supplementary Data 7). A gene ontology (GO) analysis revealed that nearly all modules were enriched for genes involved in specific biological processes, including modules enriched for

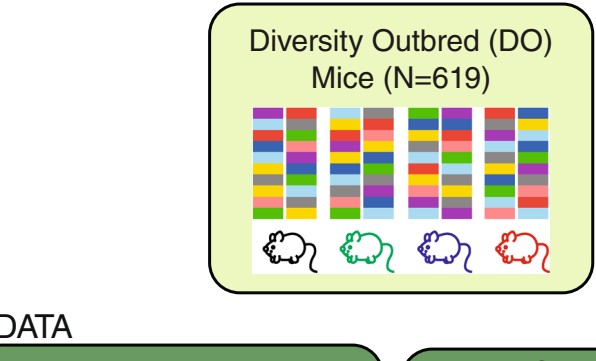

**DATA**

**Phenotypes**

55 traits covering five categories:
1) Geometry
2) Biomechanics
3) Microarchitecture
4) Marrow adiposity
5) Histomorphometry

**Genotypes**

- GigaMuga array gentoypes
(N=619 mice, ~110K markers)
- Reconstructed haplotypes

**Transcriptomics**

- RNA-seq on cortical bone
(N=192; 96 males/96 females)
- scRNA-seq on bone marrow-derived
stromal cells (N=5 mice, 7,092 cells)

**ANALYSES**

**Informing GWAS**

Co-expression networks
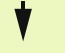
Bayesian network analysis
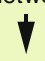
Bone-associated nodes
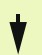
Bone-associated nodes located in
GWAS regions and regulated
by a colocalizing eQTL

**Genetic Analysis of Bone
Strength and Related Traits**

Genome scans
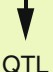
QTL

Merge anlysis; eQTL; Missense SNPs
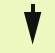
Candidate Causal Genes

**Fig. 1 Resource overview.** An overview of the resource including data generated and analyses performed. GWAS genome-wide association study, QTL quantitative trait loci, eQTL expression quantitative trait loci, SNP single-nucleotide polymorphism.

processes specific to bone cells (osteoblasts or osteoclasts) (Supplementary Data 8).

We next sought to infer causal interactions between genes in each module, and then use this information to identify genes likely involved in regulatory processes relevant to bone and the regulation of BMD. To do so, we generated Bayesian networks for each co-expression module, allowing us to model directed gene–gene relationships based on conditional independence. Bayesian networks allowed us to model causal links between co-expressed (and likely co-regulated) genes.

We hypothesized that key genes involved in bone regulatory processes would play central roles in bone networks and, thus, be more highly connected in the Bayesian networks. In order to test this hypothesis, we generated a list of genes implicated in processes known to impact bone or bone cells ("known bone gene" list ($N = 1291$; Supplementary Data 9; see "Methods"

section). The GWAS loci referenced in this study were enriched in human homologs of genes in the "known bone gene" list, relative to the set of protein-coding genes in the genome (OR = 1.35, $P = 1.45 \times 10^{-7}$). Across the three network sets (combined, male and female), we found that genes with putative roles in bone regulatory processes were more highly connected than all other genes ($P = 3.5 \times 10^{-4}$, $P = 1.7 \times 10^{-2}$, and $P = 2.9 \times 10^{-5}$ for combined, male, and female network sets, respectively), indicating the structures of the Bayesian networks were not random with respect to connectivity.

To discover genes potentially responsible for GWAS associations, we identified bone-associated nodes (BANs). BANs were defined as genes connected in our Bayesian networks with more genes in the "known bone gene" list than would be expected by chance[29–32]. The analysis identified 1370 genes with evidence ($P_{\text{nominal}} \leq 0.05$) of being a BAN (i.e., sharing network

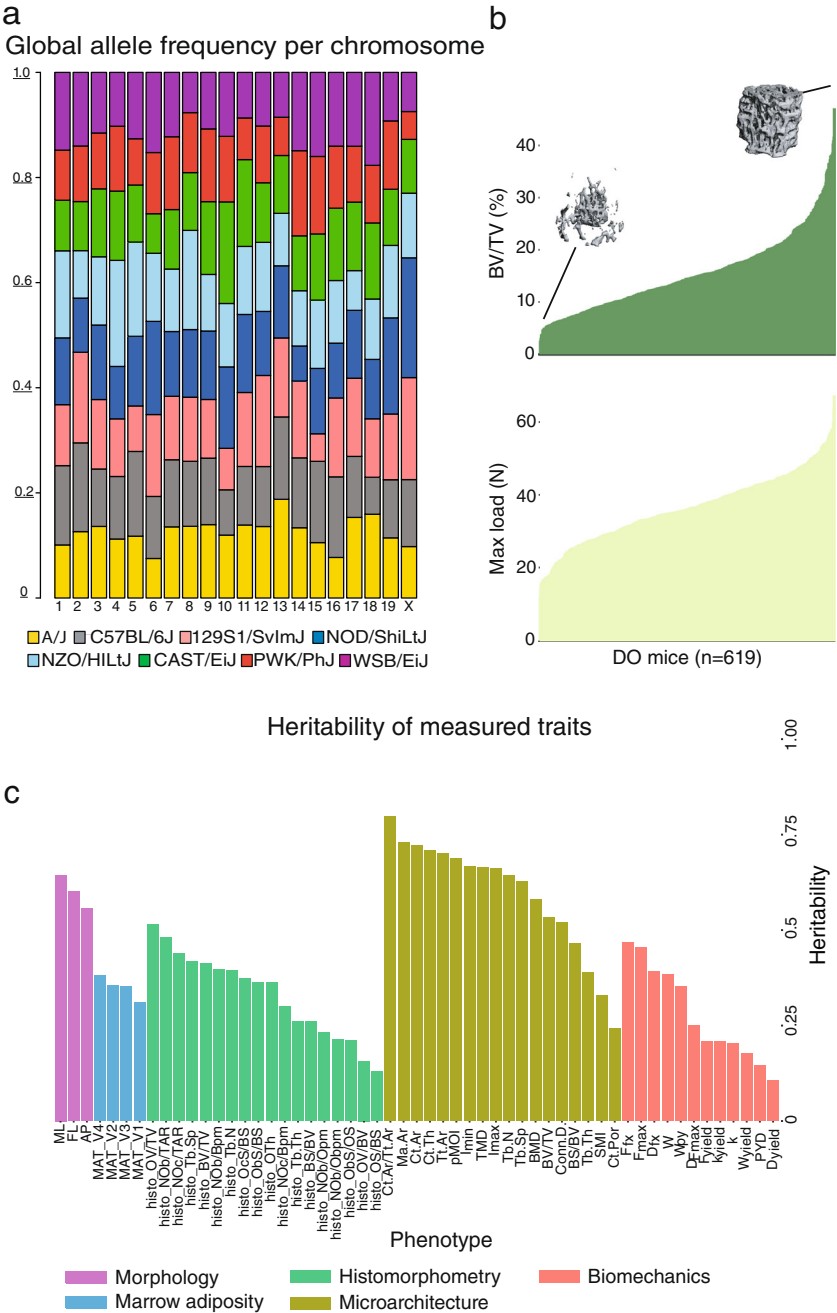

**Fig. 2 Characterization of the experimental Diversity Outbred cohort. a** Allele frequency per chromosome, across the DO cohort. Intervals represent the eight DO founder strains: A/J (yellow), C57BL/6J (gray), 129S1/SvlmJ (beige), NOD/ShiLtJ (dark blue), NZO/HILtJ (light blue), CAST/EiJ (green), PWK/PhJ (red), and WSB/EiJ (purple). **b** Bone volume fraction and max load across the DO cohort. Insets are microCT images representing low and high bone volume fraction (BV/TV). **c** Heritability of each bone trait. Phenotypes are colored by phenotypic category: morphology (purple), marrow adiposity (light blue), histomorphometry (green), microarchitecture (olive), and biomechanics (beige). Abbreviations for phenotypes are available in Supplementary Data 1.

connections with genes known to participate in a bone regulatory process) (Supplementary Data 10).

**Using BANs to inform human BMD GWAS.** We reasoned that the BAN list was enriched for causal BMD GWAS genes. In fact, of the 1370 BANs, 1173 had human homologs and 688 of those were within 1 Mbp of one of the 1161 BMD GWAS lead SNPs identified in the refs. [3,5]. This represents an enrichment of BANs within GWAS loci (+/− 1 Mbp of GWAS SNP), relative to the number of protein-coding genes within GWAS loci (OR = 1.26, $P = 9.49 \times 10^{-5}$).

However, a gene being a BAN is likely not strong evidence, by itself, that a particular gene is causal for a BMD GWAS association. Therefore, to provide additional evidence connecting BMD-associated variants to the regulation of BANs, we identified local eQTL for each BAN homolog in 48 human non-bone samples using the Genotype-Tissue Expression (GTEx) project[10,33,34]. Our rationale for using GTEx was that while these data do not include information on bone tissues or bone cells, a

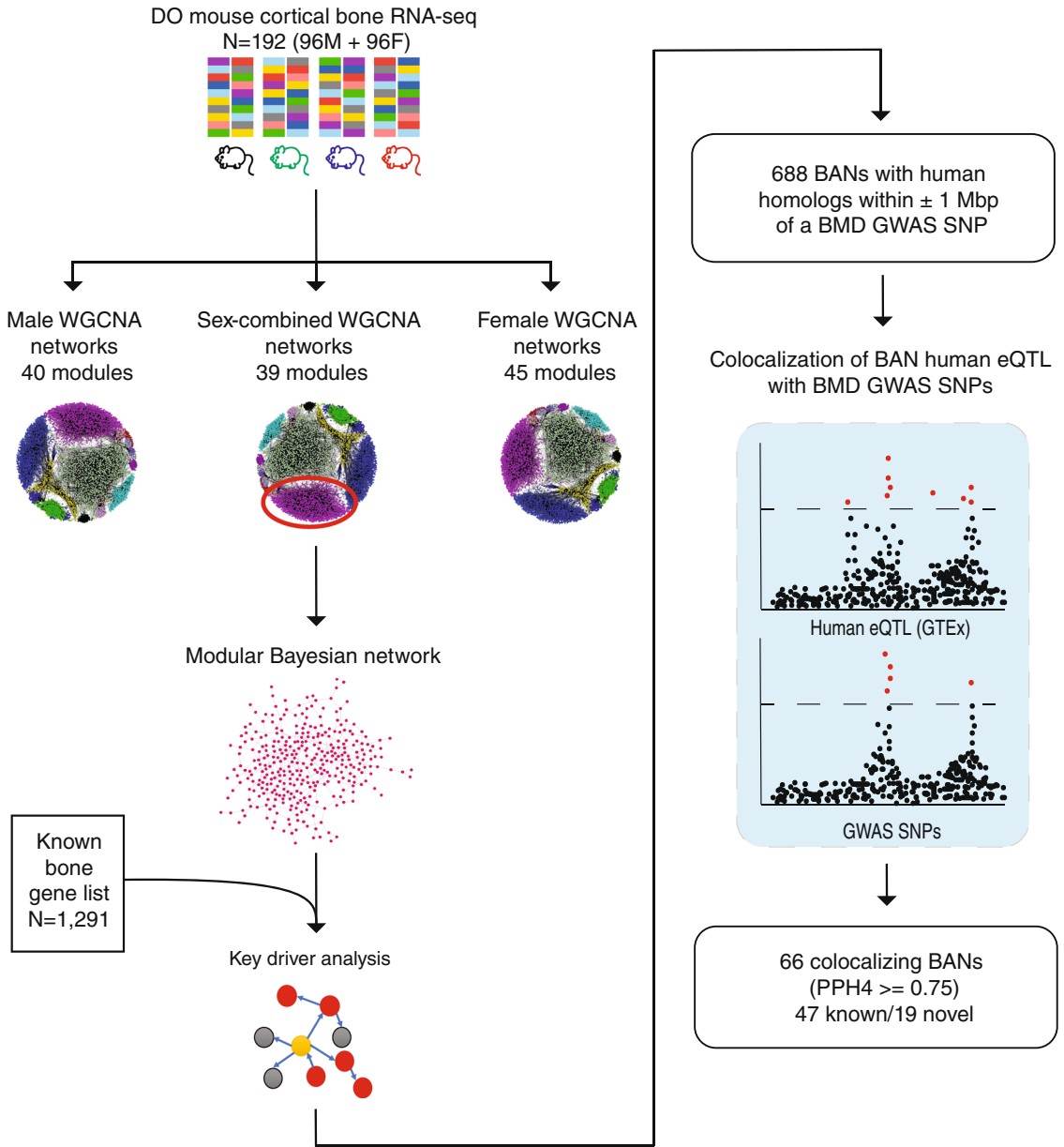

**Fig. 3 Overview of the network approach used to identify genes potentially responsible for BMD GWAS loci.** Three WGCNA networks (124 total modules) were constructed from RNA-seq data on cortical bone in the DO ($N = 192$). A Bayesian network was then learned for each module. We performed key driver analysis on each Bayesian network to identify BANs, by identifying nodes (genes) that were more connected to more known bone genes than was expected by chance. We colocalized GTEx human eQTL for each BAN with GWAS BMD SNPs to identify potentially causal genes at BMD GWAS loci. For the key driver analysis, the yellow node indicates the queried gene, red nodes indicate known bone genes, and gray nodes indicate non-bone genes. DO diversity outbred, WGCNA weighted gene co-expression network analysis, BAN bone-associated nodes, eQTL expression quantitative trait loci, BMD bone mineral density, GWAS genome-wide association studies, SNP single-nucleotide polymorphism, PPH4 posterior probability of colocalization, hypothesis 4.

high degree of local eQTL sharing has been observed between GTEx tissues[10,35]. This suggests that a colocalizing eQTL in a non-bone tissue may represent either a non-bone autonomous causal effect, or may reflect the actions of a shared eQTL that is active in bone and shared across non-bone tissues. We then tested each eQTL for colocalization (i.e., probability that the eQTL and GWAS association share a common causal variant) with their respective BMD GWAS association[3,5]. Of the 688 BANs located in proximity of a BMD GWAS locus, 66 had colocalizing eQTL (PPH4 ≥ 0.75, Supplementary Data 11, see "Methods" section) in at least one GTEx tissue (Supplementary Data 12). Of these, 47 (71.2%) were putative regulators of bone traits (based on

comparing to the known bone gene list ($N = 36$) and a literature search for genes influencing bone cell function ($N = 11$)), highlighting the ability of the approach to recover known biology. Based on overlap with the known bone gene list, this represents a highly significant enrichment of known bone genes in the list of BANs with colocalizing eQTL relative to the number of known bone genes in the list of GWAS-proximal BANs (OR = 2.53, $P = 3.09 \times 10^{-4}$). Our approach identified genes such as *SP7* (Osterix)[36], *SOST*[37,38], and *LRP5*[39–41], which play central roles in osteoblast-mediated bone formation. Genes essential to osteoclast activity, such as *TNFSF11* (RANKL)[42–45], *TNFRSF11A* (RANK)[46,47], and *SLC4A2*[48] were also identified. Nineteen

(28.8%) genes were not previously implicated in the regulation of bone traits.

One of the advantages of the network approach is the ability to identify potentially causal genes, and provide insight into how they may impact BMD based on their module memberships and network connections. For example, the cyan module in the female network (cyan_F) harbored many of the known BANs that influence BMD through a role in osteoclasts (the GO term "osteoclast differentiation" was highly enriched $P = 2.8 \times 10^{-15}$ in the cyan_F module) (Supplementary Data 8). Three of the nineteen novel BANs with colocalizing eQTL (Supplementary Data 12), *ATP6V1A*, *PRKCH*, and *AMZ1*, were members of the cyan module in the female network. Based on their cyan module memberships it is likely they play a role in osteoclasts. *ATP6V1A* is a subunit of the vacuolar ATPase V1 domain[49]. The vacuolar ATPase plays a central role in the ability of osteoclasts to acidify matrix and resorb bone, though *ATP6V1A* itself (which encodes an individual subunit) has not been directly connected to the regulation of BMD[49]. *PRKCH* encodes the eta isoform of protein kinase C and is highly expressed in osteoclasts[50]. *AMZ1* is a zinc metalloprotease and is relatively highly expressed in osteoclasts, and is highly expressed in macrophages, which are osteoclast precursors[50].

Next, we focused on two of the novel BANs with colocalizing eQTL, *SERTAD4* (GTEx Adipose Subcutaneous; coloc PPH4 = 0.77; PPH4/PPH3 = 7.9) and *GLT8D2* (GTEx Pituitary; coloc PPH4 = 0.88; PPH4/PPH3 = 13.4). Both genes were members of the royalblue module in the male network (royalblue_M). The function of *SERTAD4* (SERTA domain-containing protein 4) is unclear, though proteins with SERTA domains have been linked to cell cycle progression and chromatin remodeling[51]. *GLT8D2* (glycosyltransferase 8 domain containing 2) is a glycosyltransferase linked to nonalcoholic fatty liver disease[52]. In the DO, the eigengene of the royalblue_M module was significantly correlated with several traits, including trabecular number (Tb.N; rho $= -0.26$; $P = 9.5 \times 10^{-3}$) and separation (Tb.Sp; rho $= 0.27$; $P = 7.1 \times 10^{-3}$), among others (Supplementary Data 13). The royalblue_M module was enriched for genes involved in processes relevant to osteoblasts such as "extracellular matrix" ($P = 8.4 \times 10^{-19}$), "endochondral bone growth" ($P = 5.7 \times 10^{-4}$), "ossification" ($P = 8.9 \times 10^{-4}$), and "negative regulation of osteoblast differentiation" ($P = 0.04$) (Supplementary Data 8). Additionally, *Sertad4* and *Glt8d2* were connected, in their local (3-step) Bayesian networks, to well-known regulators of osteoblast/osteocyte biology (such as *Wnt16*[53], *Postn*[54,55], and *Col12a1*[56] for *Sertad4* and *Pappa2*[57], *Pax1*[57,58], and *Tnn*[59] for *Glt8d2*) (Fig. 4a, b). *Sertad4* and *Glt8d2* were strongly expressed in calvarial osteoblasts with expression increasing ($P < 2.2 \times 10^{-16}$ and $P = 6.4 \times 10^{-10}$, respectively) throughout the course of differentiation (Fig. 4c). To further investigate their expression in osteoblasts, we generated single-cell RNA-seq (scRNA-seq) data on mouse bone marrow-derived stromal cells exposed to osteogenic differentiation media in vitro from our mouse cohort ($N = 5$ mice (four females, one male), 7092 cells, Supplementary Data 14, Supplemental Fig. 2). Clusters of cell-types were grouped into mesenchymal progenitors, preadipocytes/adipocytes, osteoblasts, osteocytes, and non-osteogenic cells based on the expression of genes defining each cell-type (Supplementary Data 15). *Sertad4* was expressed across multiple cell-types, with its highest expression in a specific cluster (cluster 9) of mesenchymal progenitor cells and lower levels of expression in osteocytes (cluster 10) (Fig. 4d and Supplemental Fig. 3). *Glt8d2* was expressed in a relatively small number of cells in both progenitor and mature osteoblast populations (Fig. 4d and Supplemental Fig. 3).

Finally, we analyzed data from the International Mouse Phenotyping Consortium (IMPC) for *Glt8d2*[60]. After controlling for body weight, there was a significant ($P = 1.5 \times 10^{-3}$) increase in BMD in male *Glt8d2*$^{-/-}$ and no effect ($P = 0.88$) in female *Glt8d2*$^{-/-}$ mice (sex interaction $P = 6.9 \times 10^{-3}$) (Fig. 4e). These data were consistent with the direction of effect predicted by the human *GLT8D2* eQTL and eBMD GWAS locus, where the effect allele of the lead eBMD SNP (rs2722176) was associated with increased *GLT8D2* expression and decreased BMD. Together, these data suggest that *SERTAD4* and *GLT8D2* are causal for their respective BMD GWAS associations, and they likely impact BMD through a role in modulating osteoblast-centric processes.

**Identification of QTLs for strength-related traits in the DO.** The other key limitation of human genetic studies of osteoporosis has been the strict focus on BMD, though many other aspects of bone influence its strength. To directly address this limitation using the DO, we performed GWAS for 55 complex skeletal traits. This analysis identified 28 genome-wide significant (permutation-derived $P < 0.05$) QTLs for 20 traits mapping to ten different loci (defined as QTL with peaks within a 1.5 Mbp interval) (Table 1 and Supplemental Fig. 4). These data are presented interactively in a web-based tool (http://qtlviewer.uvadcos.io/). Of the ten loci, four impacted a single trait (e.g., medial–lateral femoral width (ML) QTL on Chr2@145.4 Mbp), while the other six impacted more than one trait (e.g., cortical bone morphology traits, cortical tissue mineral density (TMD), and cortical porosity (Ct.Por) QTL on Chr1@155 Mbp). The 95% confidence intervals (CIs) for the 21 autosomal associations ranged from 615 Kbp to 5.4 Mbp with a median of 1.4 Mbp.

**Overlap with human BMD GWAS.** We anticipated that the genetic analysis of bone strength traits in DO mice would uncover novel biology not captured by human BMD GWAS. To evaluate this prediction, we identified overlaps between the ten identified mouse loci and human BMD GWAS associations[3,5]. Of the ten mouse loci, the human syntenic regions (Supplementary Data 16) for six (60%) contained at least one independent GWAS association (Supplemental Fig. 5). We calculated the number expected by chance by randomly selecting ten human regions (of the same size) 1000 times, followed by identifying overlaps. Six overlaps corresponded to the 57th percentile of the null distribution.

**Identification of potentially causal genes.** For each locus, we defined the causal gene search space as the widest confidence interval given all QTL start and end positions ±250 Kbp. We then used a previously described approach, merge analysis, to fine-map QTL and identify likely causal genes (Fig. 5)[61]. Merge analysis was performed by imputing all known variants from the genome sequences of the eight founders onto haplotype reconstructions for each DO mouse, and then performing single variant association tests. We focused on variants in the top 15% of each merge analysis as those are most likely to be causal[61].

We next identified missense variants that were top merge analysis variants common to all QTL in a locus. We identified seven missense variants in locus 1, and eight missense variants in locus 3 (Table 1). Of the seven missense variants in locus 1, three (rs243472661, rs253446415, and rs33686629) were predicted to be deleterious by SIFT. They are all variants in the uncharacterized protein coding gene *BC034090*. In locus 3, three (rs250291032, rs215406048, and rs30914256) were predicted to be deleterious by SIFT (Supplementary Data 17). These variants were located in myeloid leukemia factor 1 (*Mlf1*), Iqcj and Schip1

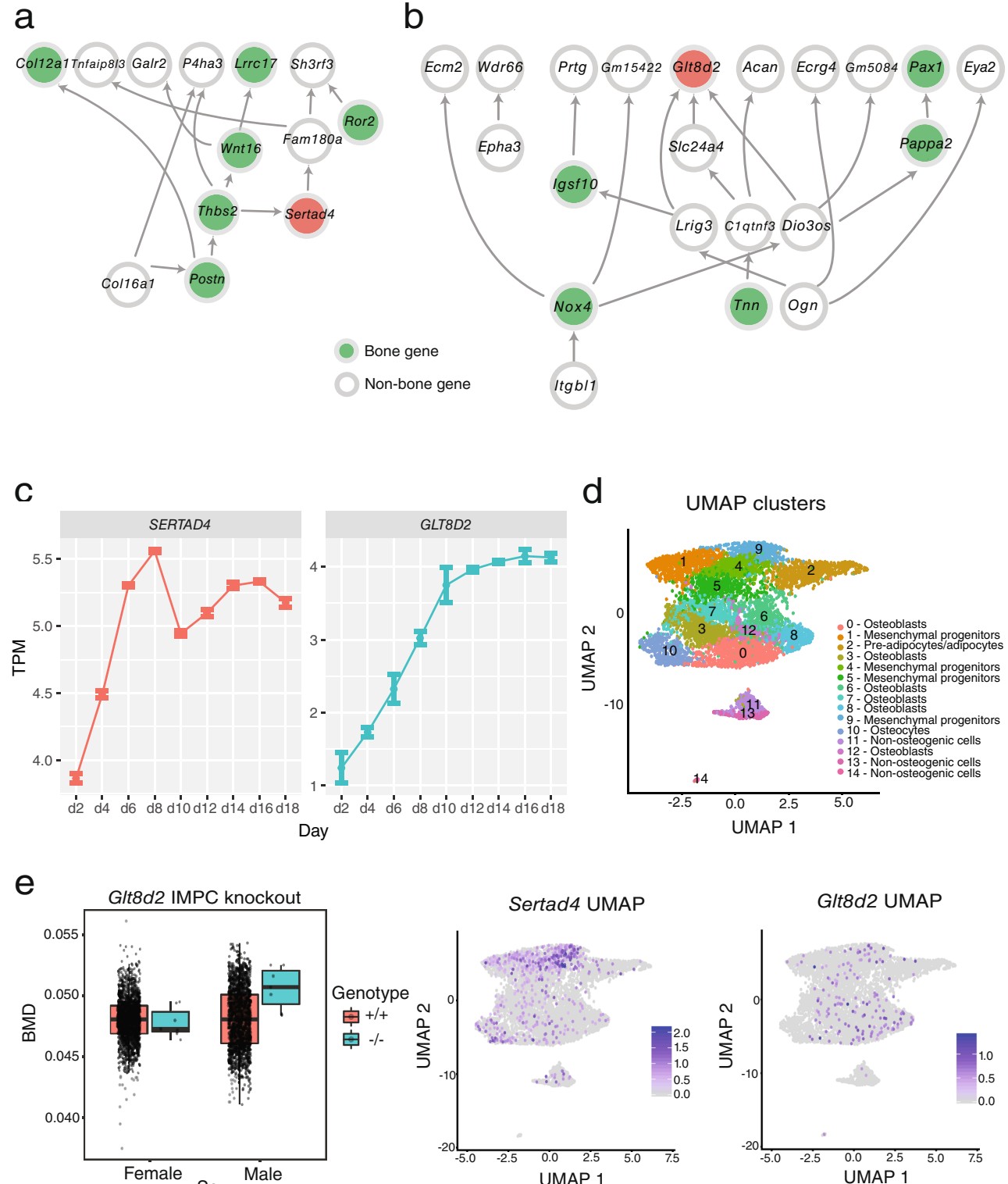

**Fig. 4 Identifying *SERTAD4* and *GLT8D2* as putative regulators of BMD. a** Local 3-step neighborhood around *Sertad4*. Known bone genes highlighted in green. *Sertad4* highlighted in red. **b** Local 3-step neighborhood around *Glt8d2*. Known bone genes highlighted in green. *Glt8d2* highlighted in red. **c** Expression of *Sertad4* and *Glt8d2* in calvarial osteoblasts. For each time point, $N = 3$ independent biological replicates were examined. Error bars represent the standard error of the mean. TPM transcripts per million. **d** Single-cell RNA-seq expression data. Each point represents a cell ($N = 7092$ cells). The top panel shows UMAP clusters and their corresponding cell-type. The bottom two panels show the expression of *Sertad4* and *Glt8d2*. The color scale indicates normalized gene expression value. **e** Bone mineral density in *Glt8d2* knockout mice from the IMPC. $N = 7$ females and $N = 7$ males for *Glt8d2*$^{-/-}$ mice, $N = 1466$ females and $N = 1477$ males for *Glt8d2*$^{+/+}$ mice. Boxplots indicate the median (middle line), the 25th and 75th percentiles (box) and the whiskers extend to 1.5 * IQR. Colors indicate genotype.

**Table 1 QTL identified for complex skeletal traits in the DO.**

| Locus | Trait | LOD | Chr. | Position (Mbp) | 95% CI (Mbp) | # missense variants | Genes with colocalizing eQTL |
|---|---|---|---|---|---|---|---|
| 1 | TMD | 23.9 | 1 | 155.1 | 154.8–155.6 | 7 | *Ier5* |
| 1 | Ma.Ar | 12.8 | 1 | 155.3 | 155.1–155.7 | 7 | *Ier5, Qsox1* |
| 1 | Tt.Ar | 11.5 | 1 | 155.2 | 155.1–156.2 | 7 | *Ier5, Qsox1* |
| 1 | Ct.Por | 11.4 | 1 | 155.4 | 155.1–156.4 | 7 | *Ier5, Qsox1* |
| 1 | ML | 10 | 1 | 155.4 | 155.1–155.7 | 7 | *Ier5, Qsox1* |
| 1 | pMOI | 8.8 | 1 | 155.1 | 154.8–158.2 | 7 | *Ier5, Qsox1* |
| 1 | Ct.Ar/Tt.Ar | 8.5 | 1 | 155.3 | 154.3–155.7 | 7 | *Ier5, Qsox1* |
| 1 | Imax | 8.3 | 1 | 155.1 | 155.1–158.2 | 7 | *Ier5, Qsox1* |
| 2 | ML | 7.9 | 2 | 145.4 | 144.1–145.6 | – | – |
| 3 | Ma.Ar | 8.8 | 3 | 68.1 | 66.6–70 | 8 | *Mfsd1, Il12a, Gm17641, 1110032F04Rik* |
| 4 | Ma.Ar | 8 | 4 | 114.6 | 113–118.4 | – | – |
| 4 | Tt.Ar | 8.2 | 4 | 114.6 | 113.6–114.8 | – | – |
| 5 | Ct.Ar/Tt.Ar | 8.1 | 4 | 127.7 | 125.4–128.1 | – | *Csf3r, Gm12946, Clspn, Ncdn, Gm12941, Zmym6, Gm25600* |
| 6 | BMD | 7.8 | 8 | 103.5 | 102.7–104.4 | – | – |
| 7 | TMD | 14.6 | 10 | 23.5 | 23.1–24.6 | – | – |
| 7 | W | 13.6 | 10 | 24.3 | 23.5–24.6 | – | – |
| 7 | Wpy | 11.9 | 10 | 23.8 | 23.5–25.3 | – | – |
| 7 | Dfx | 10.7 | 10 | 23.7 | 23.3–24.6 | – | – |
| 7 | DFmax | 9.4 | 10 | 23.7 | 21.8–25.2 | – | *C920009B18Rik* |
| 8 | Fmax | 8.8 | 16 | 23.3 | 22.3–23.4 | – | – |
| 8 | Ffx | 8.2 | 16 | 23.1 | 22.6–23.4 | – | – |
| 9 | Ct.Ar | 13.5 | X | 59.4 | 58.4–71.2 | – | – |
| 9 | Imax | 11 | X | 59.5 | 58.4–69.6 | – | – |
| 9 | pMOI | 10.4 | X | 59.4 | 58.4–61.4 | – | – |
| 9 | Imin | 8.4 | X | 59.5 | 57.3–61.2 | – | *Zic3* |
| 10 | Ct.Th | 9.9 | X | 73.4 | 58.4–74.1 | – | – |
| 10 | Tb.Sp | 8.6 | X | 73.8 | 72.7–77.5 | – | *Pls3* |
| 10 | Tb.N | 7.9 | X | 74 | 72.7–76.8 | – | *Fundc2, Cmc4, Pls3* |

fusion protein (*Iqschfp*), and Retinoic acid receptor responder 1 (*Rarres1*), respectively.

We next used the cortical bone RNA-seq data to map 10,399 local eQTL in our DO mouse cohort (Supplementary Data 18). Of these, 174 local eQTL regulated genes located within bone trait QTL. To identify colocalizing eQTL, we identified trait QTL/eQTL pairs whose top merge analysis variants overlapped. This analysis identified 18 genes with colocalizing eQTL in six QTL loci (Table 1).

**Characterization of a QTL on Chromosome 1 influencing bone morphology.** Locus 1 (Chr1) influenced cortical bone morphology (medullary area (Ma.Ar), total cross sectional area (Tt.Ar), medial–lateral femoral width (ML), polar moment of inertia (pMOI), cortical bone area fraction (Ct.Ar/Tt.Ar), and maximum moment of inertia ($I_{max}$)), tissue mineral density (TMD), and cortical porosity (Ct.Por) (Fig. 6a). We focused on this locus due to its strong effect size and the identification of candidate genes (*Ier5*, *Qsox1*, and *BC034090*) (Table 1). Additionally, we had previously measured ML in an independent cohort of DO mice ($N = 577$; 154 males/423 females) from earlier generations (generations G10 and G11) and a QTL scan of those data uncovered the presence of a similar QTL on Chr1[62] (Supplemental Fig. 6, see "Methods" section). The identification of this locus across two different DO cohorts (which differed in generations, diets, and ages) provided robust replication justifying further analysis.

The traits mapping to this locus fell into two phenotypic groups, those influencing different aspects of cross-sectional size (e.g., ML and Tt.Ar) and TMD/cortical porosity. We suspected that locus 1 QTL underlying these two groups were distinct, and that QTL for traits within the same phenotypic group were linked. This hypothesis was further supported by the observation that correlations among the size traits were strong and cross-

sectional size traits were not correlated with TMD or porosity (Supplementary Data 4). Therefore, we next tested if the locus affected all traits or was due to multiple linked QTL. The non-reference alleles of the top merge analysis variants for each QTL were private to WSB/EiJ. To test if these variants explained all QTL, we performed the same association scans for each trait, but included the genotype of the lead ML QTL variant (rs50769082; 155.46 Mbp; ML was used as a proxy for all the cortical morphology traits) as an additive covariate. This led to the ablation of all QTL except for TMD which remained significant (Fig. 6b). We then repeated the analysis using the lead TMD QTL variant (rs248974780; 155.06 Mbp) as an additive covariate (Fig. 6c). This led to the ablation of all QTLs. These results supported the presence of at least two loci both driven by WSB/EiJ alleles, one influencing cortical bone morphology and Ct.Por and the other influencing TMD, as well as possibly influencing cortical bone morphology and Ct.Por.

**Qsox1 is responsible for the effect of locus 1 on cortical bone morphology.** Given the importance of bone morphology to strength, we sought to focus on identifying the gene(s) underlying locus 1 and impacting cortical bone morphology. We re-evaluated candidate genes in light of the evidence for two distinct QTL. Immediate Early Response 5 (*Ier5*) and quiescin sulfhydryl oxidase 1 (*Qsox1*) were identified as candidates based on the DO mouse eQTL analysis and *BC034090* as a candidate based on missense variants (Table 1). Interestingly, *Ier5* and *Qsox1* eQTL colocalized with all QTL, except the TMD QTL, where only *Ier5* colocalized, providing additional support for two distinct loci (Table 1 and Fig. 7a). We cannot exclude the involvement of the missense variants in *BC034090*; however, without direct evidence that they impacted *BC034090* function, we put more emphasis on the eQTL. As a result, based on its colocalizing eQTL and known

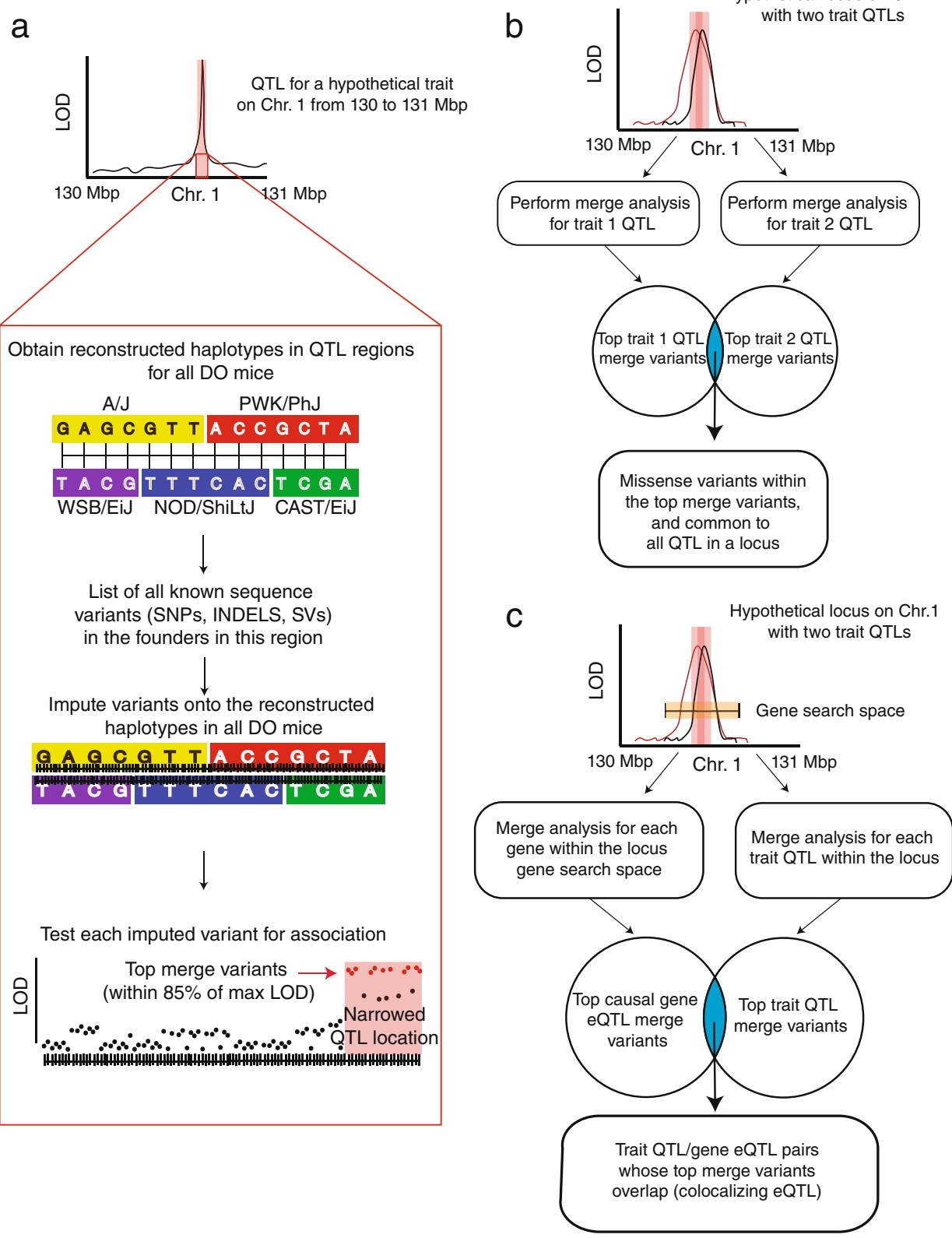

**Fig. 5 Overview of our approach to QTL fine-mapping. a** Overview of merge analysis. LOD logarithm of the odds, QTL quantitative trait loci, DO diversity outbred, SNP single-nucleotide polymorphism, INDEL insertion–deletion, SV structural variant. **b** Overview of merge analysis as performed for the identification of missense variants. **c** Overview of merge analysis as performed for the identification of colocalizing trait QTL/gene eQTL within a locus. The pink columns around the QTL in each association plot represent the QTL 95% confidence intervals. The yellow box in **c** represents the gene search space for a locus, defined as the region within ±250 Kbp around the outer boundaries of the 95% confidence intervals within a locus. eQTL expression quantitative trait loci.

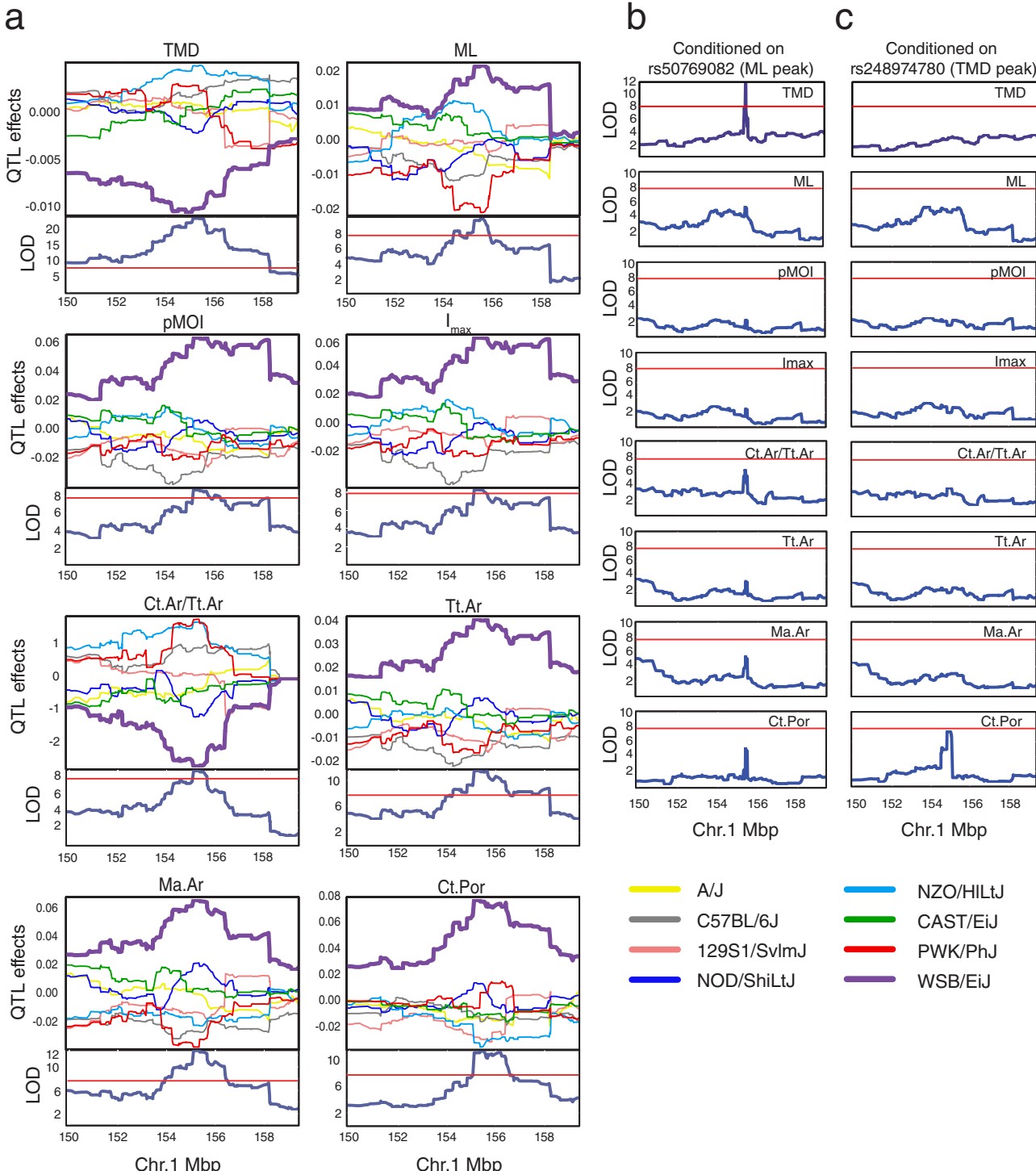

**Fig. 6 QTL (locus 1) on chromosome 1. a** For each plot, the top panel shows allele effects for the DO founders for each of the eight QTL (quantitative trait loci) across an interval on chromosome 1 (Mbp, colors correspond to the founder allele in the legend). Bottom panels show each respective QTL scan. The red horizontal lines represent LOD (logarithm of the odds) score thresholds (genome-wide $P < = 0.05$). Colors indicate founder mouse strains: A/J (yellow), C57BL/6J (gray), 129S1/SvImJ (beige), NOD/ShiLtJ (dark blue), NZO/HILtJ (light blue), CAST/EiJ (green), PWK/PhJ (red), and WSB/EiJ (purple). **b** QTL scans across the same interval as **a**, after conditioning on rs50769082. **c** QTL scans after conditioning on rs248974780. Phenotype abbreviations: TMD tissue mineral density, ML medial–lateral femoral width, pMOI polar moment of intertia, Imax maximum moment of inertia, Ct.Ar/Tt. Ar bone area fraction, Tt.Ar total area, Ma.Ar medullary area, Ct.Por cortical porosity.

biological function (see below), we predicted that *Qsox1* was at least partially responsible for locus 1.

QSOX1 is the only known secreted catalyst of disulfide bond formation and a regulator of extracellular matrix integrity[63]. It has not been previously linked to skeletal development. We found

that *Qsox1* was highly expressed in calvarial osteoblasts and its expression decreased ($P = 6.4 \times 10^{-6}$) during differentiation (Fig. 7b). In scRNA-seq on bone marrow-derived stromal cells exposed to osteogenic differentiation media in vitro, we observed *Qsox1* expression in all osteogenic cells with its highest expression

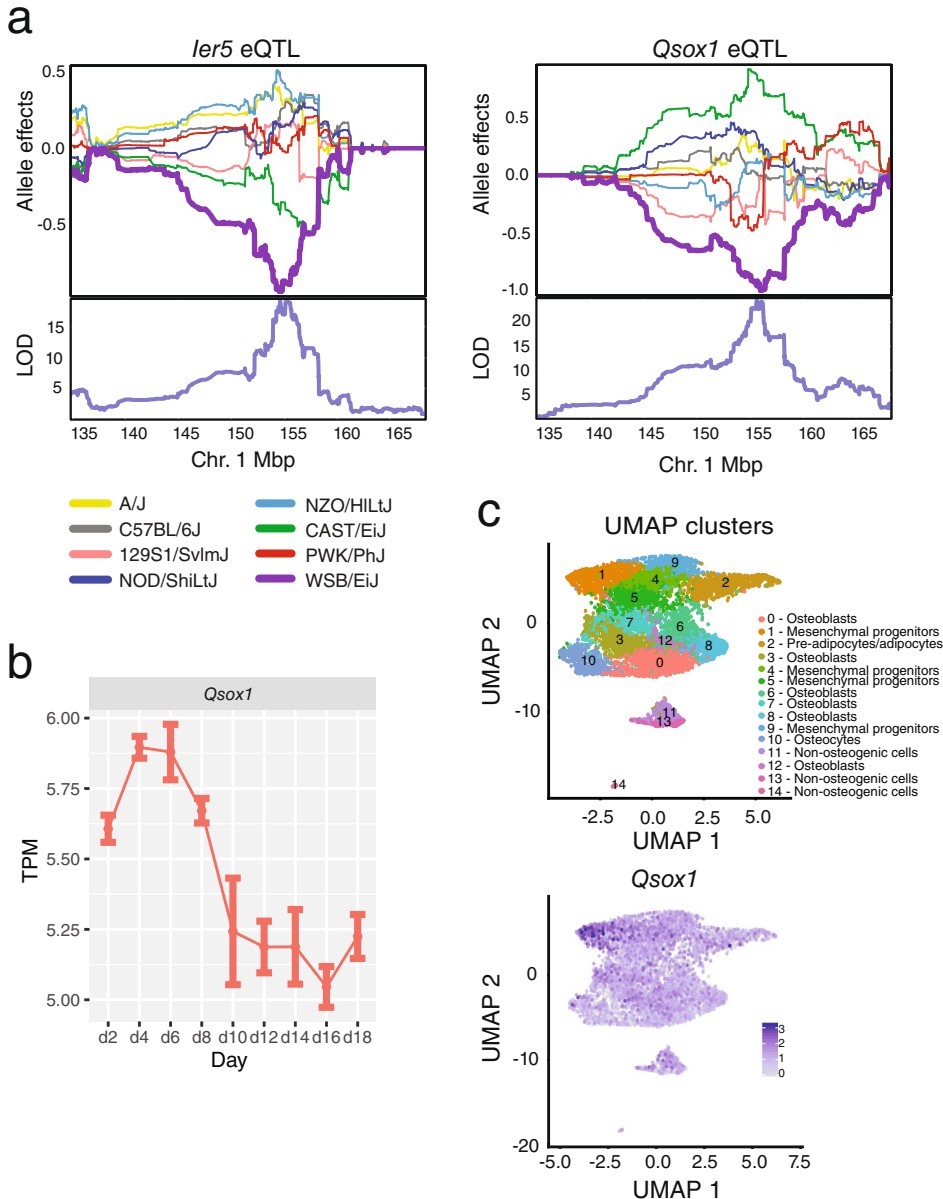

**Fig. 7 Characterization of *Qsox1*. a** The top panel shows allele effects for the DO founders for *Ier5* and *Qsox1* expression an interval on chromosome 1 (Mbp, colors correspond to the founder allele in the legend). *Y*-axis units are best linear unbiased predictors (BLUPs). Bottom panels show each respective QTL scan. LOD (logarithm of the odds) score threshold for autosomal eQTL is 10.89 (alpha = 0.05). **b** *Qsox1* expression in calvarial osteoblasts. For each time point, $N = 3$ independent biological replicates were examined. Error bars represent the standard error of the mean. TPM transcripts per million. **c** Single-cell RNA-seq expression data. Each point represents a cell ($N = 7092$ cells). The top panel shows UMAP clusters and their corresponding cell-type. The bottom panel shows the expression of *Qsox1*. The color scale indicates normalized gene expression value.

seen in a cluster of mesenchymal progenitors defined by genes involved in skeletal development such as *Grem2*, *Lmna*, and *Prrx2* (cluster 1) (Supplementary Data 19 and Fig. 7c). Additionally, in the DO cortical bone RNA-seq data, *Qsox1* was highly co-expressed with many key regulators of skeletal development and osteoblast activity (e.g., *Runx2*; rho = 0.48, $P \leq 2.2 \times 10^{-16}$, *Lrp5*; rho = 0.41, $P = 6.2 \times 10^{-9}$).

To directly test the role of *Qsox1*, we used CRISPR/Cas9 to generate five different *Qsox1* mutant mice. We generated five different mutant lines harboring unique mutations, including two 1-bp frameshifts, a 171-bp in-frame deletion of the QSOX1 catalytic domain, and two large deletions (756 and 1347 bp) spanning most of the entire first exon of *Qsox1* (Fig. 8a, Supplementary Data 20 and 21). All five mutations abolished QSOX1 activity in serum (Fig. 8b).

Given the uniform lack of QSOX1 activity, we combined phenotypic data from all lines to evaluate the effect of QSOX1 deficiency on bone. We hypothesized based on the genetic and DO mouse eQTL data, that QSOX1 deficiency would increase all traits mapping to locus 1, except TMD. Consistent with this prediction, ML was increased overall ($P = 1.8 \times 10^{-9}$), and in male ($P = 5.6 \times 10^{-7}$) and female ($P = 3.5 \times 10^{-3}$) mice as a function of *Qsox1* mutant genotype (Fig. 8c). Also consistent with the genetic data, we observed no difference in other gross morphological traits including anterior–posterior femoral width (AP) ($P = 0.31$) (Fig. 8d) and femoral length (FL) ($P = 0.64$) (Fig. 8e). We next focused on male *Qsox1*$^{+/+}$ and *Qsox1*$^{-/-}$ mice and used microCT to measure other bone parameters. We observed increased pMOI ($P = 0.02$) (Fig. 8f), Imax ($P = 0.009$)

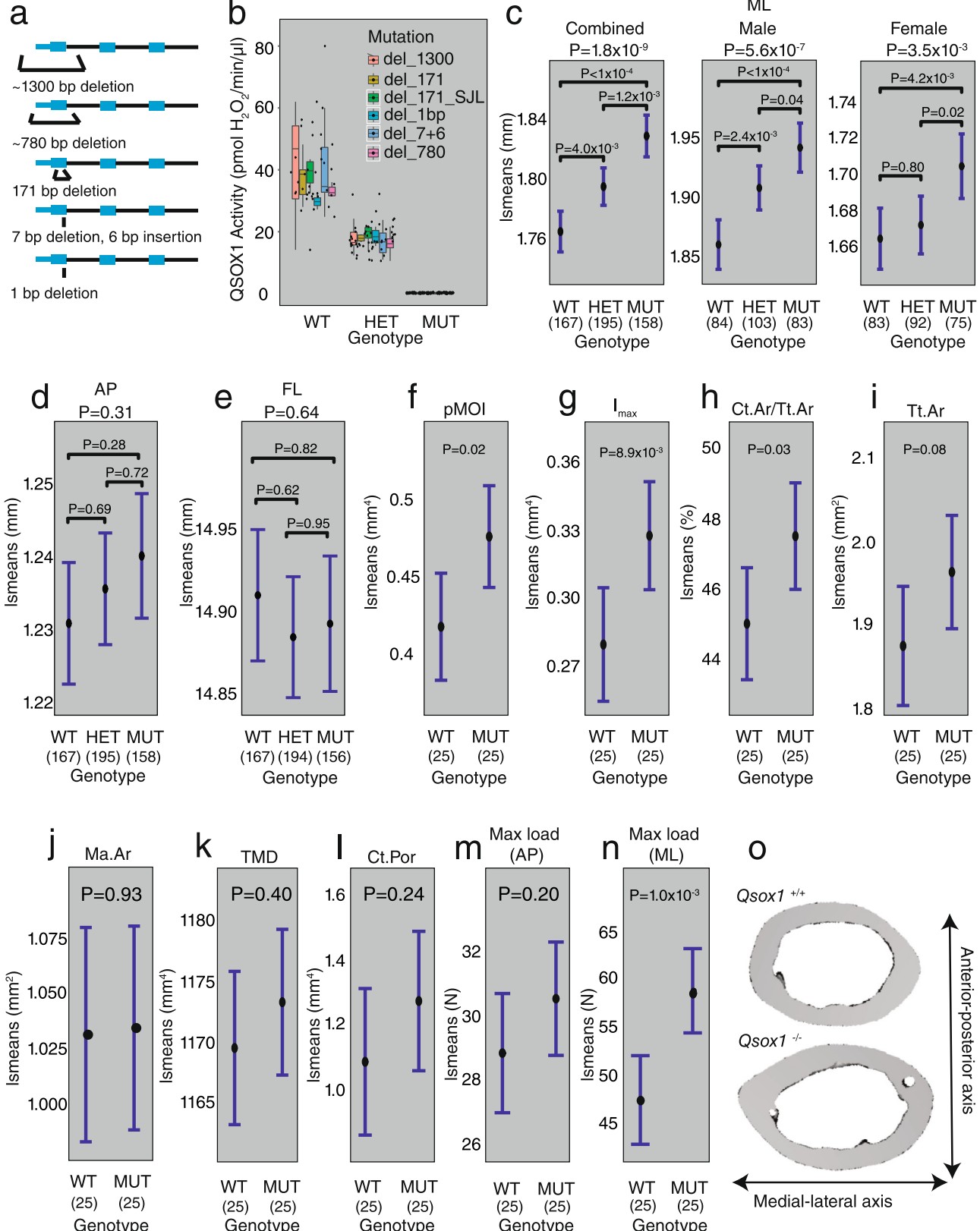

(Fig. 8g), and Ct.Ar/Tt.Ar ($P = 0.031$) (Fig. 8h). Total area (Tt.Ar) (Fig. 8i) was increased, but the difference was only suggestive ($P = 0.08$). Medullary area (Ma.Ar, $P = 0.93$) was not different (Fig. 8j). We observed no change in TMD ($P = 0.40$) (Fig. 8k). We also observed no difference in cortical porosity (Ct.Por) ($P = 0.24$) (Fig. 8l).

Given the strength of locus 1 on bone morphology and its association with biomechanical strength, we were surprised the locus did not impact femoral strength. Typically, in four-point bending assays, the force is applied along the AP axis. We replicated this in femurs from $Qsox1^{+/+}$ and $Qsox1^{-/-}$ mice and saw no significant impact on strength ($P = 0.20$) (Fig. 8m).

**Fig. 8 *Qsox1* is responsible for several chromosome 1 QTL. a** Representative image of the *Qsox1* knockout mutations. **b** QSOX1 activity assay in serum. Data is grouped by mouse genotype. Boxplots indicate the median (middle line), the 25th and 75th percentiles (box) and the whiskers extend to 1.5 * IQR. Colors indicate mutation type. **c–e** Femoral morphology in *Qsox1* mutant mice. *P*-values above plots are ANOVA *P*-values for the genotype term, while *P*-values in the plots are contrast *P*-values, adjusted for multiple comparisons. The center points of the plots represent the least-squares mean, while the error bars represent the confidence intervals at a confidence level of 0.95. **f–l** microCT measurements of chromosome 1 QTL phenotypes in *Qsox1* knockout mice. *P*-values in the plots are contrast *P*-values, adjusted for multiple comparisons. The center points of the plots represent the least-squares mean, while the error bars represent the confidence intervals at a confidence level of 0.95. **m** Bone strength (max load, $F_{max}$) in the AP orientation, measured via four-point bending. *P*-values in the plots are contrast *P*-values, adjusted for multiple comparisons. The center points of the plot represent the least-squares mean, while the error bars represent the confidence intervals at a confidence level of 0.95. **n** Bone strength (max load, $F_{max}$) in the ML orientation, measured via four-point bending. *P*-values in the plots are contrast *P*-values, adjusted for multiple comparisons. The center points of the plot represent the least-squares mean, while the error bars represent the confidence intervals at a confidence level of 0.95. **o** Representative microCT images of the effect of *Qsox1* on bone size. All error bars represent confidence intervals at a 95% confidence level. ML medial-lateral femoral width, AP anterior-posterior femoral width, FL femoral length, pMOI polar moment of inertia, Imax maximum moment of inertia, Ct.Ar/Tt.Ar bone area fraction, Tt.Ar total area, Ma.Ar medullary area, TMD tissue mineral density, Ct.Por cortical porosity.

However, when we tested femurs by applying the force along the ML axis, we observed a significant increase in strength in *Qsox1*$^{-/-}$ femurs ($P = 1.0 \times 10^{-3}$) (Fig. 8n). Overall, these data demonstrate that absence of QSOX1 activity leads to increased cortical bone accrual specifically along the ML axis (Fig. 8o).

## Discussion

Human GWASs for BMD have identified over 1000 loci. However, progress in causal gene discovery has been slow and BMD explains only part of the variance in bone strength and the risk of fracture[7]. The goal of this study was to demonstrate that systems genetics in DO mice can help address these limitations. Towards this goal, we used cortical bone RNA-seq data in the DO and a network-based approach to identify 66 genes likely causal for BMD GWAS loci. Nineteen of the 66 were novel. We provide further evidence supporting the causality of two of these genes, SERTAD4 and GLT8D2. Furthermore, GWAS in the DO identified 28 QTLs for a wide-range of strength associated traits. From these data, *Qsox1* was identified as a genetic determinant of cortical bone mass and strength. These data highlight the power of systems genetics in the DO and demonstrate the utility of mouse genetics to inform human GWAS and bone biology.

To inform BMD GWAS, we generated Bayesian networks for cortical bone and used them to identify BANs. Our analysis was similar to key driver analyses[29–31], where the focus has often been on identifying genes with strong evidence ($P_{adj} < 0.05$) of playing central roles in networks. In contrast, we used BAN analysis as a way to rank genes based on the likelihood ($P_{nominal} \leq 0.05$) that they are involved in a biological process important to bone (based on network connections to genes known to play a role in bone biology). We then identified genes most likely to be responsible for BMD GWAS associations by identifying BANs regulated by human eQTL that colocalize with BMD GWAS loci. Together, a gene being both a BAN in a GWAS locus and having a colocalizing eQTL is strong support of causality. This is supported by the observation that ~71% of the 66 BANs with colocalizing eQTL were putative regulators of bone traits, based on a literature review and overlap with the "known bone gene" list.

One advantage of our network approach was the ability to not only identify causal genes, but use network information to predict the cell-type through which these genes are likely acting. We demonstrate this idea by investigating the two novel BANs with colocalizing human eQTL from the royalblue_M module. The royalblue_M module was enriched in genes involved in bone formation and ossification, suggesting the module as a whole and its individual members were involved in osteoblast-driven processes. This prediction was supported by the role of genes in osteoblasts that were directly connected to *Sertad4* and *Glt8d2*,

the expression of the two genes in osteoblasts, and for *Glt8d2*, its regulation of BMD in vivo. Little is known regarding the specific biological processes that are likely impacted by *Sertad4* and *Glt8d2* in osteoblasts; however, it will be possible to utilize this information in future experiments designed to investigate their specific molecular functions. For example, *Sertad4* was connected to *Wnt16*, *Ror2*, and *Postn* all of which play roles in various aspects of osteoblast/osteocyte function. Wnt signaling is a major driver of osteoblast-mediated bone formation and skeletal development[64]. Interestingly, *Wnt16* and *Ror2* play central roles in canonical (*Wnt16*) and non-canonical (*Ror2* in the *Wnt5a*/*Ror2* pathway) Wnt signaling[65] and have been shown to physically interact in chondrocytes[66]. *Postn* has also been shown to influence Wnt signaling[66,67]. These data suggest a possible role for *Sertad4* in Wnt signaling.

Despite their clinical importance, we know little about the genetics of bone traits other than BMD. Here, we set out to address this knowledge gap. Using the DO, we identified 28 QTL for a wide-range of complex bone traits. The QTL were mapped at high-resolution; most had 95% CIs < 1 Mbp[18]. This precision, coupled with merge and eQTL analyses in DO mice, allowed us to identify a small number of candidate genes for many loci. Overlap of existing human BMD GWAS association and mouse loci was no more than what would be expected by chance, suggesting that our approach has highlighted biological processes impacting bone that are independent of those with the largest effects on BMD. This new knowledge has the potential to lead to novel pathways, which could be targeted therapeutically to increase bone strength. Future studies extending the work presented here will lead to the identification of additional genes, and further our understanding of the genetics of a broad range of complex skeletal traits.

Using multiple approaches, we identified *Qsox1* as responsible for at least part of the effect of the locus on Chr. 1 impacting bone morphology. We use the term "at least part" because it is clear that the Chr. 1 locus is complex. Using ML width as a proxy for all the bone morphology traits mapping to Chr. 1, the replacement of a single WSB/EiJ allele was associated with an increase in ML of 0.064 mm. Based on this, if *Qsox1* was fully responsible for the Chr. 1 locus we would expect at least an ML increase of 0.128 mm in *Qsox1* knockout mice; however, the observed difference was 0.064 mm (50% of the expected difference). This could be due to differences in the effect of *Qsox1* deletion in the DO compared to the SJL x B6 background of the *Qsox1* knockout or to additional QTL in the Chr. 1 locus. The latter is supported by our identification of at least two QTL in the region. Further work will be needed to fully dissect this locus.

Disulfide bonds are critical to the structure and function of numerous proteins[68]. Most disulfide bonds are formed in the endoplasmic reticulum[69]; however, the discovery of QSOX1

demonstrated that disulfide bonds in proteins can be formed extracellularly[63]. Ilani et al.[63] demonstrated that fibroblasts deficient in QSOX1 had a decrease in the number of disulfide bonds in matrix proteins. Moreover, the matrix formed by these cells was defective in supporting cell–matrix adhesion and lacked incorporation of the alpha-4 isoform of laminin. QSOX1 has also been associated with perturbation of the extracellular matrix in the context of cancer and tumor invasiveness[70,71]. It is unclear at this point how QSOX1 influences cortical bone mass; however, it likely involves modulation of the extracellular matrix.

In summary, we have used a systems genetics analysis in DO mice to inform human GWAS and identify genetic determinants for a wide-range of complex skeletal traits. Through the use of multiple synergistic approaches, we have expanded our understanding of the genetics of BMD and osteoporosis. This work has the potential to serve as a framework for how to use the DO, and other mouse genetic reference populations, to complement and inform human genetic studies of complex disease.

## Methods

**Diversity outbred mouse population and tissue harvesting**. A total of 619 (315 males, 304 females) Diversity Outbred (J:DO, JAX stock #0039376) mice, across 11 generations (gens. 23–33) were procured from The Jackson Laboratory at 4 weeks of age. DO mice were fed standard chow (Envigo Teklad LM-485 irradiated mouse/ rat sterilizable diet. Product # 7912). The mice were maintained on a 12-h light/12-h dark cycle, at a temperature range of 60–76 °C, with a humidity range of 20–70%. Mice were injected with calcein (30 mg/g body weight) both 7 days and 1 day prior to sacrifice. Mice were weighed and fasted overnight prior to sacrifice. Mice were sacrificed at approximately 12 weeks of age (median: 86 days, range: 76–94 days). Immediately prior to sacrifice, mice were anesthetized with isoflurane, nose-anus length was recorded and blood collected via submandibular bleeding. At sacrifice, femoral morphology (length and width) was measured with digital calipers (Mitoyuto American, Aurora, IL). Right femora were wrapped in PBS soaked gauze and stored in PBS at −20 °C. Right tibiae were stored in 70% EtOH at room temperature. Left femora were flushed of bone marrow (which was snap frozen and stored in liquid nitrogen, see below—Single-cell RNA-seq of bone marrow stromal cells exposed to osteogenic differentiation media in vitro) and were immediately homogenized in Trizol. Homogenates were stored at −80 °C. Left tibiae were stored in 10% neutral buffered formalin at 4 °C. Tail clips were collected and stored at −80 °C.

**Measurement of trabecular and cortical microarchitecture**. Right femora were scanned using a 10 μm isotropic voxel size on a desktop μCT40 (Scanco Medical AG, Brüttisellen, Switzerland), following the Journal of Bone and Mineral Research guidelines for assessment of bone microstructure in rodents[72]. Trabecular bone architecture was analyzed in the endocortical region of the distal metaphysis. Variables computed for trabecular bone regions include: bone volume, BV/TV, trabecular number, thickness, separation, connectivity density, and the structure model index, a measure of the plate versus rod-like nature of trabecular architecture. For cortical bone at the femoral midshaft, total cross-sectional area, cortical bone area, medullary area, cortical thickness, cortical porosity, and area moments of inertia about principal axes were computed.

**Biomechanical testing**. The right femur from each mouse was loaded to failure in four-point bending in the anterior to posterior direction, such that the posterior quadrant is subjected to tensile loads. The widths of the lower and upper supports of the four-point bending apparatus are 7 mm and 3 mm, respectively. Tests were conducted with a deflection rate of 0.05 mm/s using a servohydraulic materials test system (Instron Corp., Norwood, MA). The load and mid-span deflection were acquired directly at a sampling frequency of 200 Hz. Load-deflection curves were analyzed for strength (maximum load), stiffness (the slope of the initial portion of the curve), post-yield deflection, and total work. Post-yield deflection, which is a measure of ductility, is defined as the deflection at failure minus the deflection at yield. Yield is defined as a 10% reduction of stiffness relative to the initial (tangent) stiffness. Work, which is a measure of toughness, is defined as the area under the load-deflection curve. Femora were tested at room temperature and kept moist with phosphate buffered saline during all tests.

**Assessment of bone marrow adipose tissue (MAT)**. Fixed right tibiae, dissected free of soft tissues, were decalcified in EDTA for 20 days, changing the EDTA every 3–4 days and stained for lipid using a 1:1 mixture of 2% aqueous osmium tetroxide ($OsO_4$) and 5% potassium dichromate. Decalcified bones were imaged using μCT performed in water with energy of 55 kVp, an integration time of 500 ms, and a maximum isometric voxel size of 10 μm (the "high" resolution setting with a 20 mm sample holder) using a μCT35 (Scanco). To determine the position of the

MAT within the medullary canal and to determine its change in volume, the bone was overlaid. MAT was recorded in four dimensions.

**Histomorphometry**. Fixed right tibiae were sequentially dehydrated and infiltrated in graded steps with methyl methacrylate. Blocks were faced and 5 μm non-decalcified sections cut and stained with toludine blue to observe gross histology. This staining allows for the observation of osteoblast and osteoclast numbers, amount of unmineralized osteoid and the presence of mineralized bone. Histo-morphometric parameters were analyzed on a computerized tablet using Osteo-measure software (Osteometrics, Atlanta, GA). Histomorphometric measurements were made on a fixed region just below the growth plate corresponding to the primary spongiosa.

**Bulk RNA isolation, sequencing, and quantification**. We isolated RNA from a randomly chosen subset ($N = 192$, 96/sex) of the available mice at the time (mice number 1–417), constrained to have an equal number of male and female mice. Total RNA was isolated from marrow-depleted homogenates of the left femora, using the mirVana™ miRNA Isolation Kit (Life Technologies, Carlsbad, CA). Total RNA-Seq libraries were constructed using Illumina TruSeq Stranded Total RNA HT sample prep kits. Samples were sequenced to an average of 39 million $2 \times 75$ bp paired-end reads (total RNA-seq) on an Illumina NextSeq500 sequencer in the University of Virginia Center for Public Health Genomics Genome Sciences Laboratory (GSL). A custom bioinformatics pipeline was used to quantify RNA-seq data. Briefly, RNA-seq FASTQ files were quality controlled using FASTQC (version 0.11.5)[73] and MultiQC (version 1.0.dev0)[74], aligned to the mm10 genome assembly with HISAT2 (version 2.0.5)[75], and quantified with Stringtie (version 1.3.3)[76]. Read count information was then extracted with a Python script provided by the Stringtie website (prepDE.py). Finally, we filtered our gene set to include genes that had more than six reads, and more than 0.1 transcripts per million (TPM), in more than 38 samples (20% of all samples). This filtration resulted in 23,648 genes remaining from an initial set of 53,801 genes. (Note that most of these genes were defined by StringTie internally as genes, but indicate loci—contiguous regions on the genome where the exons of transcripts overlap). Sequencing data is available on GEO at accession code GSE152708.

**Bulk RNA differential expression analyses**. RNA-seq data were subjected to a variance stabilizing transformation using the DESeq2 (version 1.20.0) R package[77], and the 500 most variable genes were used to calculate the principal components using the PCA function from the FactoMineR (version 2.4) R package[78]. For visualization, age was binarized into "high" and "low", with "low" defined as age equal to, or less than, the median age at sacrifice (85 days) and "high" defined as age higher than 85 days. Differential expression was then performed using DESeq2, for both sex and bone strength (max load). For differential expression based on sex, we used a design formula of ~batch + age + sex. For bone strength, we binarized bone strength into "high" and "low" for each sex independently, using the median bone strength value for each sex (35.66 and 37.42 for males and females, respectively). Differential expression was performed using the following design formula: ~sex + batch + age + bone strength. Log2 fold changes for both differential expression analyses were then shrunken using the lfcShrink function in DESeq2, using the adaptive t prior shrinkage estimator from the apeglm (version 1.4.2) R package[79].

**Mouse genotyping**. DNA was collected from mouse tails from all 619 DO mice, using the PureLink Genomic DNA mini kit (Invitrogen). DNA was used for genotyping with the GigaMUGA array[19] by Neogen Genomics (GeneSeek; Lincoln, NE). Genotyping reports were pre-processed for use with the qtl2 (version 0.20) R package[80,81], and genotypes were encoded using directions and scripts from (kbroman.org/qtl2/pages/prep_do_data.html). Quality control was performed using the Argyle (version 0.2.2) R package[82], where samples were filtered to contain no more than 5% no calls and 50% heterozygous calls. Samples that failed QC were re-genotyped. Furthermore, genotyping markers were filtered to contain only tier 1 and tier 2 markers. Markers that did not uniquely map to the genome were also removed. Finally, a qualitative threshold for the maximum number of no calls and a minimum number of homozygous calls was used to filter markers.

We calculated genotype and allele probabilities, as well as kinship matrices using the qtl2 R package. Genotype probabilities were calculated using a hidden Markov model with an assumed genotyping error probability of 0.002, using the Carter–Falconer map function. Genotype probabilities were then reduced to allele probabilities, and allele probabilities were used to calculate kinship matrices, using the "leave one chromosome out" (LOCO) parameter. Kinship matrices were also calculated using the "overall" parameter for heritability calculations.

Further quality control was then performed[83], which led to the removal of several hundred more markers that had greater than 5% genotyping errors, after which genotype and allele probabilities and kinship matrices were recalculated. After the aforementioned successive marker filtration, 109,427 markers remained, out of 143,259 initial genotyping markers. As another metric for quality control, we calculated the frequencies of the eight founder genotypes of the DO.

**WGCNA network construction**. Gene counts, as obtained above, were pruned to remove genes that had fewer than ten reads in more than 90% of samples. Genes not located on the autosomes or X chromosome were also removed. This led to the retention of 23,335 out of 23,648 genes. Variance-stabilizing transformation (DeSeq2[77]) was applied, followed by RNA-seq batch correction using sex and age at sacrifice in days as covariates (sex was not included as a covariate in the sex-specific networks), using ComBat (sva (version 3.30.0) R package[84]). We then used the WGCNA (version 1.68) R package to generate signed co-expression networks with a soft thresholding power of 4 (power = 5 for male networks)[85,86]. We used the blockwiseModules function to construct networks with a merge cut height of 0.15 and minimum module size of 30. WGCNA networks had 39, 45, and 40 modules for the sex-combined, female, and male networks, respectively.

**Bayesian network learning**. Bayesian networks for each WGCNA module were learned with the bnlearn (version 4.5) R package[87]. Specifically, expression data for genes within a WGCNA module were obtained as above (WGCNA network construction), and these data were used to learn the structure of the underlying Bayesian network using the Max-Min Hill Climbing algorithm (function mmhc in bnlearn).

**Construction of the "known bone gene" list**. We constructed a list of bone genes using GO terms and the Mouse Genome Informatics (MGI) database[88,89]. Using AmiGO2, we downloaded GO terms for "osteo*", "bone" and "ossif*", using all three GO domains (cellular component, biological process and molecular function), without consideration of GO evidence codes[90]. The resulting GO terms were pruned to remove some terms that were not related to bone function or regulation. We then used the MGI Human and Mouse Homology data table to convert human genes to their mouse homologs. We also downloaded human and mouse genes which had the terms "osteoporosis", "bone mineral density", "osteoblast", "osteoclast", and "osteocyte", from MGI's Human Mouse: Disease Connection (HMDC) database. Human genes were converted to their mouse counterparts as above. GO and MGI derived genes were merged and duplicates were removed. Finally, we removed genes that were not expressed in our dataset. That is to say, they were not considered in generating the WGCNA modules or Bayesian networks.

**Bone associated node (BAN) analysis**. We used a custom script that utilized the igraph (version 1.2.4.1) R package to perform BAN analysis[91]. Briefly, within a Bayesian network underlying a WGCNA module, we counted the number of neighbors for each gene, based on a neighborhood step size of 3. Neighborhood sizes also included the gene itself. BANs were defined as genes that were more highly connected to bone genes than would be expected by chance. We merged all genes from all Bayesian networks together in a matrix, and removed genes that were unconnected or only connected to 1 neighbor (neighborhood size ≤ 2). We then pruned all genes whose neighborhood size was greater than 1 standard deviation less than the mean neighborhood size across all modules. These pruning steps resulted in 13,009/17,264, 11,861/16,446 and 11,877/17,042 genes remaining for the full, male and female Bayesian networks, respectively.

Then, for each gene, we calculated if they were more connected to bone genes in our bone list (see construction of bone list above) than expected by chance using the hypergeometric distribution (phyper, R stats (version 3.5.1) package). The arguments were as follows: $q$: (number of genes in neighborhood that are also bone genes) – 1; $m$: total number of bone genes in our bone gene set; $n$: (number of genes in networks prior to pruning) – $m$; $k$: neighborhood size of the respective gene; lower.tail = false.

**GWAS-eQTL colocalization**. We converted mouse genes with evidence of being a BAN ($P ≤ 0.05$) to their human homologs using the MGI homolog data table. If the human homolog was within 1 Mbp of a GWAS association, we obtained all eQTL associations within +/− 200 kb of the GWAS association in all 48 tissues of version 7 of the Genotype-Tissue Expression project (GTEx). These eQTL variants were colocalized with the GWAS variants, using the coloc.abf function from the R coloc (version 3.2.1) package[92]. This returned posterior probabilities (PP) for five hypotheses:

H0: No association with either trait.
H1: Association with trait 1, not with trait 2.
H2: Association with trait 2, not with trait 1.
H3: Association with traits 1 and 2, two independent SNPs.
H4: Association with traits 1 and 2, one shared SNP.

Genes were considered colocalizing if PPH4 ≥ 0.75.

**Gene ontology**. GO analysis for WGCNA modules was performed for each individual module using the topGO (version 2.32.0) package in R[93]. Enrichment tests were performed for the "Molecular Function", "Biological Process" and "Cellular Component" ontologies, using all genes in the network. Enrichment was performed using the "classic" algorithm with Fisher's exact test. $P$-values were not corrected for multiple testing.

**Assessing the expression of *Glt8d2* and *Sertad4* in publicly available bone cell data**. We used bioGPS expression data from GEO with the accession code of GSE10246 to assay the expression of *Sertad4*, *Glt8d2*, and *Qsox1* in osteoblasts[50]. We also downloaded the data from GEO with the accession code GSE54461 to query expression in primary calvarial osteoblasts.

**Analysis of BMD data on *Glt8d2*⁻/⁻ mice from the IMPC**. The International Mouse Knockout Consortium[60] and the IMPC[94] have generated and phenotyped mice harboring null alleles for *Glt8d2* (*Glt8d2*^tm1a(KOMP)Wtsi, *Glt8d2*⁻/⁻) ($N = 7$ females and $N = 7$ males). Phenotypes for the appropriate controls (C57BL/6) were also collected ($N = 1466$ females and $N = 1477$ males). A description of the battery of phenotypes collected on mutants can be found at (https://www.mousephenotype.org/impress/PipelineInfo?id=4). The mice were 14 weeks of age at DEXA scanning and both sexes were included. We downloaded raw BMD, body weight and metadata for *Glt8d2* mutants from the IMPC webportal [https://www.mousephenotype.org/data/charts?accession=MGI:1922032&allele_accession_id=MGI:4364018&pipeline_stable_id=MGP_001&procedure_stable_id=IMPC_DXA_001¶meter_stable_-id=IMPC_DXA_004_001&zygosity=homozygote&phenotyping_center=WTSI]. These data were analyzed using PhenStat (version 2.18.1) R package[95]. PhenStat was developed to analyze data generated by the IMPC in which a large number of wild-type controls are phenotyped across a wide-time range in batches, and experimental mutant animals are tested in small groups interspersed among wild-type batches. We used the Mixed Model framework in PhenStat to analyze BMD data. The mixed model framework starts with a full model (with fixed effects of genotype, sex, genotype × sex and weight and batch as a random effect) and ends with final reduced model and genotype effect evaluation procedures[95,96].

**QTL mapping**. Phenotypes that notably deviated from normality were $\log_{10}$-transformed (the MAT phenotypes as well as PYD and $W_{py}$ were transformed after a constant of 1 was added). Then, QTL mapping with a single-QTL model was performed via a linear mixed model using the scan1 function of the qtl2 R package. A kinship matrix as calculated by the "leave one chromosome out" method was included. Mapping covariates were sex, age at sacrifice in days, bodyweight, and DO mouse generation. Peaks were then identified with a minimum LOD score of 4 and a peak drop of 1.5 LODs. To identify significant QTL peaks, we permuted each phenotype scan 1000 times (using the scan1perm function of the qtl2 package) with the same mapping covariates as above, and calculated the significance threshold for each phenotype at a 5% significance level. Heritability for the phenotypes was calculated using the est_herit function of the qtl2 R package, using the same covariates as above, but with a kinship matrix that was calculated using the "overall" argument.

**DO eQTL mapping**. Variance stabilizing transformation was applied to gene read counts from above using the DESeq2 R package, followed by quantile-based inverse Normal transformation[97]. Then, hidden determinants of gene expression were calculated from these transformed counts, using Probabilistic Estimation of Expression Residuals (PEER (version 1.3))[98]. Forty-eight PEER factors were calculated using no intercept or covariates. Sex and the 48 PEER covariates were used as mapping covariates, and eQTL mapping was performed using the scan1 function, as above. To calculate a LOD score threshold, we randomly chose 50 genes and permuted them 1000 times, as above. Since all genes were transformed to conform to the same distribution, we found that using 50 was sufficient. Thresholds were set as the highest permuted LOD score each for autosomal chromosomes and the X-chromosome (10.89 and 11.55 LODs, respectively). Finally, we identified peaks as above, and defined eQTL as peaks that exceeded the LOD threshold and were no more than 1 Mbp away from their respective transcript's start site, as defined by the Stringtie output.

**Merge analysis**. We performed merge analysis, a previously published approach, using the SNP-association methods in the qtl2 R package[61,81]. For each DO mouse QTL or eQTL peak, we imputed all variants within the 95% confidence interval of a peak, and tested each variant for association with the respective trait. This was performed using the scan1snps function of the qtl2 R package, with the same mapping covariates for QTL or eQTL, respectively. Then, we identified "top" variants by taking variants that were within 85% of the maximum SNP association's LOD score. For conditional analyses using a variant, we performed the same QTL scan as above, but included the genotype of the respective SNP as an additive mapping covariate, encoding it as a 0, 0.5, or 1, for homozygous alternative, heterozygous, or homozygous reference, respectively.

**BMD-GWAS overlap**. To identify BMD GWAS loci that overlapped with our DO mouse associations, we defined a mouse association locus as the widest confidence interval given all QTL start and end CI positions mapping to each locus. We then used the UCSC liftOver tool (https://genome.ucsc.edu/cgi-bin/hgLiftOver)[99] (minimum ratio of bases that must remap = 0.1, minimum hit size in query = 100,000) to convert the loci from mm10 to their syntenic hg19 positions. We then took all genome-wide significant SNPs ($P ≤ 5 \times 10^{-8}$) from the Morris et al. GWAS for eBMD and the Estrada et al. GWAS for FNBMD and LSBMD, and identified

variants that overlapped with the syntenic mouse loci (GenomicRanges (version 1.32.7) R package[100]).

**SIFT annotations**. SIFT annotations for merge analysis missense variants were queried using Ensembl's Variant Effect Predictor tool (https://useast.ensembl.org/Tools/VEP)[101]. All options were left as default.

**Prior ML QTL mapping**. The cohorts used for the earlier QTL mapping of ML consisted of 577 Diversity Outbred mice from breeding generations G10 and G11[62]. G10 cohort mice consisted of both males and females fed a defined synthetic diet (D10001, Research Diets, New Brunswick, NJ), and were euthanized and analyzed at 12–15 weeks of age. G11 cohort mice were all females fed a defined synthetic diet (D10001, Research Diets, New Brunswick, NJ) until 6 weeks of age, and were then subsequently fed either a high-fat, cholesterol-containing (HFC) diet (20% fat, 1.25% cholesterol, and 0.5% cholic acid) or a low-fat, high protein diet (5% fat and 20.3% protein) (D12109C and D12083101, respectively, Research Diets, New Brunswick, NJ), and were euthanized and analyzed at 24–25 weeks of age. Mice were weighed and then euthanized by $CO_2$ asphyxiation followed by cervical dislocation. Carcasses were frozen at −80 °C. Subsequently, the femur was dissected and length, AP width, and ML width were measured two independent times to 0.01 mm using digital calipers. Mice were genotyped using the Mega-MUGA SNP array (GeneSeek; Lincoln, NE) designed with 77,800 SNP markers, and QTL mapping was performed as above, but with the inclusion of sex, diet, age, and weight at sacrifice as additive covariates.

**Generation of *Qsox1* mutant mice**. *Qsox1* knockout mice used in this study were generated using the CRISPR/Cas9 genome editing technique essentially as reported in Mesner et al.[102]. Briefly, Cas9 enzyme that was injected into B6SJLF2 embryos (described below) was purchased from (PNA Bio) while the guide RNA (sgRNA) was designed and synthesized as follows: the 20 nucleotide (nt) sequence that would be used to generate the sgRNA was chosen using the CRISPR design tool developed by the Zhang lab (crispr.mit.edu). The chosen sequence and its genome map position are homologous to a region in Exon 1 that is ~225 bp 3′ of the translation start site and ~20 bp 5′ of the Exon1/Intron1 boundary (Supplementary Data 22). To generate the sgRNA that would be used for injections oligonucleotides of the chosen sequence, as well as the reverse complement (Supplementary Data 22, primers 1 and 2, respectively), were synthesized such that an additional 4 nts (CACC and AAAC) were added to the 5′ ends of the oligonucleotides for cloning purposes. These oligonucleotides were annealed to each other by combining equal molar amounts heating to 90 °C for 5 min, and allowing the mixture to passively cool to room temperature. The annealed oligonucleotides were combined with BbsI digested pX330 plasmid vector (provided by the Zhang lab through Addgene; https://www.addgene.org/) and T4 DNA ligase (NEB) and subsequently used to transform Stbl3 competent bacteria (Thermo Fisher) following the manufacturer's protocols. Plasmid DNAs from selected clones were sequenced from primer 3 (Supplementary Data 22) and DNA that demonstrated accurate sequence and position of the guide were used for all downstream applications. The DNA template used in the synthesis of the sgRNA was the product of a PCR using the verified plasmid DNA and primers 4 and 5 (Supplementary Data 22). The sgRNA was synthesized via in vitro transcription (IVT) by way of the MAXIscript T7 kit (Thermo Fisher) following the manufacturer's protocol. sgRNAs were purified and concentrated using the RNeasy Plus Micro kit (Qiagen) following the manufacturer's protocol.

B6SJLF1 female mice (Jackson Laboratory) were super-ovulated and mated with B6SJLF1 males. The females were sacrificed and the fertilized eggs (B6SJLF2 embryos) were isolated from the oviducts. The fertilized eggs were co-injected with the purified Cas9 enzyme (50 ng/µl) and sgRNA (30 ng/µl) under a Leica inverted microscope equipped with Leitz micromanipulators (Leica Microsystems). Injected eggs were incubated overnight in KSOM-AA medium (Millipore Sigma). Two-cell stage embryos were implanted on the following day into the oviducts of pseudopregnant ICR female mice (Envigo). Pups were initially screened by PCR of tail DNA using primers 6 and 7 with subsequent sequencing of the resultant product from primer 8, when the PCR products suggested a relatively large deletion had occurred in at least one of the alleles (Supplementary Data 22). For those samples which indicated a small or no deletion had occurred, PCR of tail DNA using primers 9 and 10 was performed with subsequent sequencing of the resultant products from primer 11 (Supplementary Data 22). Finally, deletions were fully characterized by ligating, with T4 DNA ligase (NEB), the PCR products from either primer pairs 6/7 or 9/10 with the plasmid vector pCR 2.1 (Thermo Fisher) followed by transformation of One Shot Top 10 chemically competent cells (Thermo Fisher) following the manufacturers recommendations (Supplementary Data 22).

The resulting founder mice (see Supplementary Data 20) were mated to C57BL/6J mice (Jackson Laboratory), with CRISPR/Cas9-deletion heterozygous F1 offspring from the 1st and 2nd litters mated to generate the F2 offspring used in the study of bone related properties reported herein. In addition, mouse B (Supplementary Data 20) was subsequently mated to an SJL/J male (Jackson Laboratory), and the F2 offspring from the heterozygous F1 crosses, as outlined above, were also used in this study. All F1 and F2 mice from all deletion "strains" were genotyped using primer pairs 9/10, with the PCR products sequenced from

primer 11 for mice possessing the 7 + 6 and 1 bp deletions (Supplementary Data 22). An additional PCR using primers 6 and 7 was performed with tail DNA from mice carrying the 1347 and 756 bp deletions; the products from this 2nd PCR assisted in determining between heterozygous and homozygous deleted genotypes (Supplementary Data 22).

ML was measured for both femurs using calipers on a population of 12-week old F2 mice and ML was averaged between the two femurs. A linear model with genotype, mutation type, length, and weight was generated separately for males and females. For the sex-combined data, a sex term was also included in the model. ANOVAs were performed using the Anova function from the car (version 3.0.7) R package[103]. Lsmeans were calculated using the emmeans (version 1.4.1) R package[104]. The same procedure was performed for the AP and FL sex-combined data.

We randomly selected 50 male F2 mice (25 wt + 25 mut) from the same population, and microarchitectural phenotypes were measured as above, but on left femurs. Bone strength was measured as above but in both the AP and ML orientations. A linear model with genotype, mutation type and weight was generated, and lsmeans were calculated using the emmeans R package[104].

**Measuring Qsox1 activity in serum**. Serum was collected via submandibular bleeding from isoflurane anesthetized mice, prior to sacrifice and isolation of femurs for bone trait analysis. Blood samples were incubated at room temperature for 20–30 m followed by centrifugation at 2000 × g for 10 m at 4 °C. The supernatants were transferred to fresh tubes and centrifuged again as described above. The 2nd supernatant of each sample was separated into 50–100 µl aliquots, snap frozen on dry ice and stored at −70 °C. Only "clear" serum samples were used for determining QSOX1 activity, because pink-red colored samples had slight-moderate activity, presumably due to sulfhydryl oxidase enzymes released from lysed red blood cells.

Sulfhydryl oxidase activity was determined as outlined in Israel et al.[105] with minor modifications. Briefly, serum samples were thawed on wet ice whereupon 5 µl was used in a 200 µl final reaction volume which consisted of 50 mM $KPO_4$, pH 7.5, 1 mM EDTA (both from Sigma), 10 µM Amplex UltraRed (Thermo Fisher), 0.5% (v/v) Tween 80 (Surface-Amps, low peroxide; Thermo Fisher), 50 nM Horseradish Peroxide (Sigma), and initiated with the addition of dithiothreitol (Sigma) to 50 µM initial concentration. The reactions were monitored with the "high-sensitive dsDNA channel" of a Qubit Fluorimeter (Thermo Fisher) by measuring the fluorescence every 15–30 s for 10 m. The assay was calibrated by adding varying concentrations (0–3.2 µM) of freshly diluted $H_2O_2$ (Sigma) to the reaction mixture minus serum. Enzyme activity was expressed in units of (pmol $H_2O_2$/min/µl serum) and typically calculated within the first several minutes of the reaction for wild-type and heterozygous mutant mice. Enzyme activity was calculated during the entire 10 minutes of the reaction for homozygous mutant genotypes.

**Single-cell RNA-seq of bone marrow stromal cells exposed to osteogenic differentiation media in vitro**

*Bone marrow isolation*. The left femur was isolated and cleaned thoroughly of all muscle tissue followed by removal of its distal epiphysis. The marrow was exuded by centrifugation at 2000×g for 30 s into a sterile tube containing 35 µl freezing media (90% FBS, 10% DMSO). The marrow was then triturated six times on ice after addition of 150 µl ice cold freezing media and again after further addition of 1 ml ice cold freezing media until no visible clumps remained prior to being placed into a Mr. Frosty Freezing Container (Thermo Scientific) and stored overnight at −80 °C. Samples were transferred the following day to liquid nitrogen for long term storage.

*Bone marrow culturing*. Previously frozen bone marrow samples from 5 DO mice (mouse IDs: 12, 45, 48, 50, and 84) were thawed at 37 °C, resuspended into 5 ml bone marrow growth media (Alpha MEM, 10% FBS, 1% Pen/Strep, 0.01% Glutamax), pelleted in a Sorvall tabletop centrifuge at 212 × g for 5 min at room temperature and then subjected to red blood cell lysis by resuspending and triturating the resultant pellet into 5 ml 0.2% NaCl for 20 s, followed by addition and thorough mixing of 1.6% NaCl. Cells were pelleted again, resuspended into 1 ml bone marrow growth media, plated into one well per sample of a 48 well tissue culture plate and placed into a 37 °C, 5% $CO_2$ incubator undisturbed for 3 days post-plating, at which time the media was aspirated, cells were washed with 1 ml DPBS once and bone marrow growth media was replaced at 300 µl volume. The process was repeated through day 5 post-plating. At day 6 post-plating, cells were washed in same manner; however, we performed a standard in vitro osteoblast differentiation protocol, by replacing bone marrow growth media with 300 µl osteogenic differentiation media (Alpha MEM, 10% FBS, 1% Penicillin Streptomycin, 0.01% Glutamax, 50 mg/ml Ascorbic Acid, 1 M B-glycerophosphate, 100 µM Dexamethasome). Cells undergoing differentiation were assessed for accumulated mineralization on days 4, 6, 8, and 10 of the differentiation process as follows: IRDye 680 BoneTag Optical Probe (Li-Cor Biosciences, product #926-09374) was reconstituted according to the manufacturer's instructions. On days 3, 5, 7, and 9, 0.006 nanomoles were added to each sample. Twenty-four hours later the cells were washed with 0.5 ml DPBS (Gibco, product #14190250) and media was replaced. The cells were then placed on the Odyssey CLx Imaging System (Li-Cor Biosciences) to measure mineralization density as reflected by IRDye 680

BoneTag Optical Probe incorporation. Final values for mineralization were computed by subtracting the average number of fluorescent units recorded in designated background wells, from the number of fluorescent units recorded in the sample wells.

*RNA isolation.* The isolation procedure outlined below was inspired by[106]. Mineralized cultures were washed twice with Dulbecco's Phosphate Buffered Saline (DPBS). 0.5 ml 60 mM EDTA (pH 7.4, made in DPBS) was added for 15-min room temperature (RT) incubation. EDTA solution was aspirated and replaced for a second 15-min RT incubation. Cultures were then washed with 0.5 ml Hank's Balanced Salt Solution (HBSS) and incubated with 0.5 ml 8 mg/ml collagenase in HBSS/4 mM $CaCl_2$ for 10 min at 37 °C with shaking. Cultures were triturated 10× and incubated for an additional 20 min and 37 °C. Cultures were then transferred to a 1.5 ml Eppendorf tube, and spun at 500×*g* for 5 min at RT in a Sorvall tabletop centrifuge. Cultures were resuspended in 0.5 ml 0.25% trypsin-EDTA (Gibco, Gaithersburg, MD) and incubated for 15 min at 37 °C. Cultures were then triturated and incubated for an additional 15 min. 0.5 ml of media were added, triturated and spun at 500×*g* for 5 min at RT. Cultures were then resuspended in 0.5 ml bone marrow differentiation media and cells were counted.

*Library preparation, sequencing, and analysis.* The samples were pooled and concentrated to 800 cells/μl in sterile PBS supplemented with 0.1% BSA. The single cell suspension was loaded into a 10x Chromium Controller (10x Genomics, Pleasanton, CA, USA), aiming to capture 8000 cells, with the Single Cell 3′ v2 reagent kit, according to the manufacturer's protocol. Following GEM capturing and lysis, cDNA was amplified (13 cycles) and the manufacturer's protocol was followed to generate the sequencing library. The library was sequenced on the Illumina NextSeq500 and the raw sequencing data was processed using CellRanger toolkit (version 2.0.1). The reads were mapped to the mm10 mouse reference genome assembly using STAR (version 2.5.1b)[107]. Overall, 7188 cells were sequenced, to a mean depth of 57,717 reads per cell. Sequencing data is available on GEO at accession code GSE152806.

Analysis was performed using Seurat (version 3.1.4)[108,109]. Features detected in at least three cells where at least 200 features were detected were used. We then filtered out cells with less than 800 reads and more than 5800 reads, as well as cells with 10% or more mitochondrial reads. This resulted in 7105 remaining cells. Expression measurements were multiplied by 10,000 and log normalized, and the 3000 most variable features were identified. The data were then scaled. Cells were then scored by cell cycle markers, and these scores, as well as the percentage of mitochondrial reads, were regressed out[110]. Finally, clusters were found with a resolution of 1 and the UMAP was generated. An outlier cluster consisting of 13 cells was removed, resulting in 7092 remaining cells. Cluster cell types were manually annotated after performing differential expression analyses of the expression of genes in each cluster relative to all other clusters (Supplementary Data 15), using the Seurat FindAllMarkers function, with the only.pos = TRUE argument.

**Reporting summary**. Further information on research design is available in the Nature Research Reporting Summary linked to this article.

## Data availability

Raw genotyping data, calculated genotype and allele probabilities, and R/qtl2 cross files are available from Zenodo at 10.5281/zenodo.4265417[111]. Raw sequencing data is available from the NCBI Gene Expression Omnibus database with accession codes GSE152708 and GSE152806. Mapped DO mouse QTL and eQTL can be viewed at our web-based tool [http://qtlviewer.uvadcos.io/]. eBMD GWAS summary statistics used for this study are available from GEFOS [http://www.gefos.org/?q=content/data-release-2018], as are the FN and LS BMD GWAS summary statistics [http://www.gefos.org/?q=content/data-release-2012]. We used bioGPS expression data from GEO with the accession code of GSE10246 to assay the expression of Sertad4, Glt8d2, and Qsox1 in osteoblasts. We also downloaded the data from GEO with the accession code GSE54461 to query expression in primary calvarial osteoblasts. Glt8d2 knockout data was downloaded from the IMPC [https://www.mousephenotype.org/data/charts?accession=MGI:1922032&allele_accession_id=MGI:4364018&pipeline_stable_id=MGP_001&procedure_stable_id=IMPC_DXA_001¶meter_stable_id=IMPC_DXA_004_001&zygosity=homozygote&phenotyping_center=WTSI]. Mouse-Human homologs were obtained from MGI [http://www.informatics.jax.org/downloads/reports/HOM_MouseHumanSequence.rpt]. We also obtained data from the MGI Human-Mouse:Disease Connection database [http://www.informatics.jax.org/diseasePortal]. Gene Ontologies were obtained from AmiGO2 [http://amigo.geneontology.org/amigo]. Finally, we obtained expression data from version 7 of the Genotype-Tissue Expression project [https://gtexportal.org/home/datasets].

## Code availability

Analysis code is available on GitHub [https://github.com/basel-maher/DO_project][112].

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

## Acknowledgements

Research reported in this publication was supported in part by the National Institute of Arthritis and Musculoskeletal and Skin Diseases of the National Institutes of Health under Award Numbers AR057759 to C.J.R., M.C.H., and C.R.F., and AR077992 to C.R.F. B.M.A-B. was supported in part by a National Institutes of Health, Biomedical Data Sciences Training Grant (5T32LM012416). The authors acknowledge Wenhao Xu (University of Virginia) and the Genetically Engineered Mouse Models (GEMM) core and the University of Virginia Cancer Center Support Grant (CCSG) P30CA044579 from NCI for their support in generating *Qsox1* mutant mice. The authors also acknowledge the Yale School of Medicine Department of Orthopaedics and Rehabilitation's Histology and Histomorphometry Laboratory for all their work. We thank Matt Vincent (The Jackson Laboratory) and Gary Churchill (The Jackson Laboratory) for developing the QTL Viewer software and Neal Magee (University of Virginia) for hosting QTL Viewer on UVA servers. We thank the IMPC for accessibility to BMD data on *Glt8d2* knockout mice (www.mousephenotype.org). The data used for the analyses described in this manuscript were obtained from the IMPC Portal on 11/5/19. The Genotype-Tissue Expression (GTEx) Project was supported by the Common Fund of the Office of the Director of the National Institutes of Health, and by NCI, NHGRI, NHLBI, NIDA, NIMH, and NINDS. The data used for the analyses described in this manuscript were obtained from the GTEx Portal on 01/15/18.

## Author contributions

C.R.F., C.J.R., M.C.H., M.L.B., and S.M.T. initiated and designed the studies. B.M.A.-B., L.D.M., G.M.C., D.B., S.M.T., K.N., and E.A.F., performed the experiments. S.O.-G. provided guidance for single-cell RNA-seq experiments. D.P. provided genotypic and phenotypic data from an independent DO cohort. S.H. analyzed data from the International Mouse Phenotyping Consortium. All authors contributed to the interpretation of data/results. C.R.F. oversaw the project. C.R.F. and B.M.A.-B. wrote the manuscript with input from most of the other authors. All authors had access to the final manuscript and approved the submission of the article.

## Competing interests

The authors declare no competing interests.

## Ethics declaration

The animal protocol for the characterization of Diversity Outbred mice and the generation and characterization of *Qsox1* mutant mice was approved by the Institutional Animal Care and Use Committee (IACUC) at the University of Virginia.
