## [Peer Review File · Nature Communications]

Reviewers' Comments:

Reviewer #1:

Remarks to the Author:

In this manuscript, Al-Barghouthi et al. utilize DO mice to map and identify genes that play a role in a variety of bone-related traits. Integrating these studies with gene expression studies in mice and humans together with human GWAS data, the authors report cumulative evidence supporting the role of a number of candidate genes in mouse bone development, as well as potentially towards interpreting human GWAS data, although the link between these is less convincing. The manuscript culminates in the generation of *Qsox1* knockout mice which are used to demonstrate that this gene is required for optimal bone development and function. The manuscript is well written, the methods are well detailed, and many novel insights were discovered. It's not clear to me that all of the claims in the manuscript are strongly supported by the evidence, however it nonetheless represents an important advance in our understanding of the genetics of bone-related complex traits.

Concerns:

1. It's not clear that the statement in the abstract on line 45 and elsewhere in the paper describing the genes identified as "likely to be causal for human BMD GWAS" is fully supported by the evidence.

For example, for *Glt8d2* this claim is based on:

- a. Results from a knockout mouse that is probably unlikely to mimic the effect from a GWAS SNP/haplotype and is associated with partial lethality.
- b. An eQTL from a potentially unrelated tissue/cell type (although in this case pituitary could be relevant). Also, my understanding is that the eQTL was based on proximity to the BAN and not linked to the risk GWAS allele which means the GWAS allele may not be associated with any expression differences in *Glt8d2*.
- c. Expression within the bone marrow.

I certainly agree each of these add to the support that *Glt8d2* could be causal, but without any context of what is known about nearby genes or specific effects on expression or function based on the risk GWAS haplotype, it seems premature to claim that the causal genes have been likely identified.

2. How many BAN genes were also on the list of known bone genes? Is the 29 known bone gene enrichment on line 175 based on the GTEX eQTLs further enriched with genes on the known bone list relative to the whole list of 900 BANs with human homologs or 544 near the GWAS SNPs?

3. How much of the genome is covered by the 1,103 BMD genome SNPs \pm their 1 MB window? Are the BANs enriched for being near GWAS SNPs?

4. It would be nice if Table 1 included the tissue(s) that the eQTLs were based on.

5. The IMPC data for the *Glt8d2* KO is used to identify an increase in BMD in male mice (Fig. 2e). Can the authors comment on why the IMPC website lists no significant results for these mice for any skeletal trait including cortical BMD in the KO mice? Same question for *Qsox1* which is also in the IMPC database and has no significant skeletal phenotypes.

6. The manuscript mentions another cohort of DO mice (line 298). Has this data been published elsewhere? If not, a fuller description of this cohort and its QTL analysis would help support its use as replication of the Chr 1 loci, as well as including whether or not this cohort supported any of the other QTL loci.

Typos:

1. Line 85. Should say "heterogenous" instead of "heterozygous"??
2. Line 167. Should say "causal" instead of "casual".
3. Line 179. Should say "mediated" instead of "medicated".
4. Line 286. "identify" instead of "identifying"

Reviewer #2:

Remarks to the Author:

Al-Barghouthi et al submitted a paper evaluating bone phenotyping data obtained from Diversity Outbred (DO) mice to find new candidate genes associated with bone-disease-related traits. The authors point out that there are many associations known for links between genetic variants and osteoporotic traits. However, these associations are mainly linked to changes in bone mineral density only, not making use of other traits that could be relevant as a biomarker for bone disease. They applied a network approach to cortical bone RNA-seq data and found 46 genes with a presumably causal link to human bone disease some of them novel in the field. Candidates were validated using existing or newly generated phenotyping data. In the next step, the authors performed a GWAS in their DO cohort using a wide range of bone traits and identified *Qsox1* as new candidate gene affecting bone structural features.

The analysis addresses the very relevant disease area of osteoporosis affecting not only patients in the U.S but all over the world impacting health and quality of life, especially in aged patients. Several analyses have been published already attempting to identify candidate genes associated with human bone disease based on data generated from experimental models such as laboratory mice. These candidate genes and the genetic networks respectively physiological pathways are thought to offer new targets for the prevention and treatment of human disease.

The presented analysis comprises an impressive range of methods both regarding bone-related phenotypes in DO mice and additional mouse models used to validate the findings as well as network and genetic analysis to identify new candidate genes associated with bone disease and gain additional evidence for a link to bone metabolism.

Overall, the authors present a solid and interesting genetic analysis resulting in a small number of new candidate genes. The role of these candidate genes in bone metabolism are discussed. I have only a few questions/remarks that need clarification from my point of view:

It was not quite clear to me, which genes dropped out as candidates during the different analysis steps (also see remarks below). The author state that there are over 1000 known candidates so I also wonder why only a few new candidates were detected.

While reading the introduction and results sections I was wondering what are the founders of DO and do they show differences in bone-related parameters. Later I realized that this information is provided e.g. in Suppl. Figure 1 – could these details be presented more prominently?

The authors point out that so far analyses are mainly based on BMD data not making use of a more in-depth analysis of structural characteristics of bone presumably related to bone disease. Here, I wonder 1) what is the translational potential for changes detected in mice and the link to human clinics, and 2) are the 50 complex traits somehow inter-correlated, or do they occur in syndrome-like clusters? Were the traits treated as independent in this analysis? Could there be a kind of "hierarchy" for these traits?

There are a couple of minor remarks that may require clarification:

Line – 82 – please list the founder strains here

Line 111 – here it is stated that 55 complex traits were measured – not 50?
Line 126 – were these mice food-deprived for 24 hours? Why so long and could that affect bone marrow adiposity or other parameters?
Line 132 – what exactly does 30.8-fold “variation” mean?
Line 229 – IMPC data was used for validation – did this mutant line have further phenotypes (e.g. metabolism or X-ray)?
Line 264 ff – this statement is somehow difficult to understand and appears a bit speculative.
Line 286 – to identify instead of to “identifying”?
Line 294 – do the authors interpret effects on several complex traits as “pleiotropic”? Is this justified or is it an effect on the same feature just affecting several complex traits?
Line 493 – please specify the “standard chow” diet – manufacturer, product number, chow, or purified diets? These details should always be provided to allow the exact replication of the results.
Line 494 – see above: why were mice food-deprived for 24 hours? May that affect any of the parameters e.g. bone marrow adiposity?
Line 495 – “approximately 12 weeks” – what was the range? Why did the authors choose relatively young mice? I assume age affects bone phenotypes.
Line 565 ff – In this section, a couple of exclusion criteria were described. It would be good to know how many genes were excluded because they failed QC.
Line 583 and 589 – how many genes were removed from the analysis due to QC issues?
Line 655 – why “both mutants”? – not clear
Line 671 – why was “DO mouse generation” included as a factor – did mouse generation vary a lot?
Line 715 – specify “standard chow diet” (see above)
Line 717 – specify the HFC, low-fat, and high protein diet (see above)
Line 718 – age here 24-25 weeks – why were mice older?
Line 772 – for Qsox1 mutant mice a C57BL6J was used – Rasd1 was likely on an N background – why did you change the background? Could that have a consequence on bone phenotypes?

In addition:

1000 loci associated with bone phenotypes can be found in the literature - can this list be provided as supplemental information?

Reviewer #3:

Remarks to the Author:

Reviewer summary: In this study, Al-Barghouthi et al. measured >50 skeletal traits in a cohort (N=619) taken from the Diversity Outbred (DO) mouse population and use this to discover molecular features correlated with changes in bone morphology or function. Generating immense transcriptomic and genotype datasets, their multimodal data analysis approach identifies genes that influence skeletal phenotype in mice and potentially contribute to human skeletal trait variance.

In the transcriptome analysis, key genes were identified as those co-expressed in mouse bone tissue and strong network connectivity with genes previously shown to play a role in bone. The relevance of these genes to human bone mineral density (BMD) variation was inferred based on 1) the colocalization of human orthologs within 1Mb of a significant BMD-GWAS loci and 2) the cooccurrence of eQTLs which regulated these genes in the same region. Among those, Sertad4 and Glt8d2 were specifically examined as putative regulators of BMD, with the latter shown to impact BMD in mice. Hypotheses regarding the mechanism by which these genes impact skeletal biology were then developed, integrating network information, bulk and single-cell level transcriptome data to identify plausible cell-types and pathways.

Using the genotype data, QTLs associated with skeletal traits were identified by GWAS in mice. The overlap of these syntenic regions with BMD-associated regions in the human genome was determined. As this overlap was not statistically significant, the authors took this to indicate that the mouse QTLs

were driven by genes involved in processes influencing aspects of bone other than BMD. To identify potential causal genes associated with the QTLs, genes containing deleterious missense variants significantly associated with trait variation and a colocalizing eQTL were identified. This identified *Qsox1* as a strong causal candidate, which the authors went on to show impacts bone geometry and strength when functionally deleted in mice.

The article is creative, well written and provides possibly the largest data resource of its kind in the field of skeletal biology. As an approach, this article demonstrates how molecular genetic data from mice can complement and enrich data from human populations, guiding the search for causal genes underlying association signals and provide insight into important skeletal characteristics not measurable in humans at scale. While I am confident the data, analytical approaches and causal gene candidates will help guide future investigations into skeletal genetics, there are areas of the manuscript that should be clarified or strengthened with additional analyses and data. I have outlined these areas in the comments below.

Comments:

1) The utility of the data resources hinges on accurate meta-information, including sample relationships between datasets, inter-trait relationships and feature/gene information. There are several errors in the first supplementary figures and tables describing the phenotypes. For example:

- Supplemental figure 1 - panels b and c do not correspond with the figure legend. Panel c - it is not clear how the traits are grouped/colored, it would appear to be based on the phenotype category. Needs this information in the figure legend.

- Table S1 - abbreviations duplicated for 40 of the phenotypes (all histo_OV/TV). Also, not immediately clear how 'Number measured' and 'Number mapped' are defined. More detail required in the Table legend.

- Using gene symbols as feature identifiers (such as gene lists etc.) can be troublesome as gene names change over time with new information and commonly used software (i.e. excel) can misinterpret these character strings leading to mislabeling. It is recommended that stable, unique gene identifiers, such as Ensembl ids, be included in tables along with gene names to ensure the temporal fidelity of results across platforms. Currently table S4, S6, S7, S11 and S12 only have gene symbols as identifiers.

- Line 129-131 and Table S2-3: The p-values are very low for traits that have a relatively weak correlation with bone strength ($|r| < 0.3$, table S2). This discordance may be due to the use of Pearson correlation which relies on a number of assumptions regarding the underlying data distribution which may not be valid across such a diverse spectrum of measured traits. Spearman correlation as an alternative requires less assumptions and may be more suited in this context. This should provide a better estimate of the relationship between traits.

- One of the key strengths of this data resources is having matched phenotypic, genotype and transcriptomic data on such a large cohort. To ensure this aspect of the data can be utilized by the wider research community, it is recommended the authors include an additional table containing raw phenotyping data for each mouse, and another table to link this phenotyping information to genotype and transcriptomic data between samples. Additionally, the authors should provide more information as to the design of the RNAseq dataset e.g. How were the samples for transcriptomic analysis (N=192) chosen from the entire cohort (N=619)? It should also be explicitly state how many mice were genotyped in figure 1, results or methods.

2) Regarding the identification of 'bone associated nodes' (BANs) that have a potentially causal role in human BMD variation described in section 'Using BANs to inform human BMD GWAS'. It would be helpful if the authors could elaborate on the motivation for using both proximity to GWAS signals in syntenic regions and the colocalization of a local eQTL to prioritize BANs that are more likely to be causal. While the return of such a high fraction of genes with established role in bone seems to support such a strategy, some clarification around the hypothesis driving such a filtering approach may be warranted. i.e. Are genes that are located in proximity to a given association signal more

likely to be 'causal' if they colocalize with an eQTL?

3) In the identification of potentially causal genes associated with the trait GWAS signals in mice (Section beginning line 268), the authors impute variant information based on the GigaMUGA array and individual strain genomes, before selecting missense, trait-associated variants. As an alternative/supplementary approach, I would encourage the authors to call variants using the transcriptome sequencing data as this may be a more direct approach to mapping variants in individual mice. Further, by comparing the results of the imputation and with RNAseq variant calling across the 192 mice in the transcriptome cohort, the authors could establish the fidelity of the imputation approach and whether it is appropriate for use on the genotyped cohort (>600 mice). While this will somewhat bias the resolution to exonic regions, as the current prioritization approach selects only missense variants this should not be an issue. Further it may allow the direct comparison of phenotypes between mice with and without variants of interest, such as those with or without *Qsox1* variants.

4) The authors state that one might expect the genetic effects on bone geometry and tissue mineral density (TMD) to be distinct (Lines 303-304). Yet, this would seem to contrast a long-standing idea that bone strength is determined by both its material composition and morphology (PMID: 12584605, PMID: 16723616). This may also be discordant with the result in table S3 which shows a strong correlation (>0.8) between Medial-lateral femoral width (ML) and TMD.

While I agree the data in figure 3b and 3c do seem to indicate there are two distinct signals at this QTL, I offer a slightly different interpretation of these results. While the retention of TMD signal in Figure 3b indeed indicates TMD is independent of ML, the ablation of the ML signal when the model includes a term for TMD in Figure 3c may indicate this second locus affects both ML and TMD. It would help if the authors clarified their interpretation of this result and how their data may contrast or extend current understanding of the relationships between bone geometry and tissue mineral density.

5) Concerning the characterization of the *Qsox1* functional knockout mice. Figure 5m shows a hypothesized model of the increased cortical bone accrual specifically along the ML axis. While Figure 5c shows the ML increase, to support this model it is important to show the data that indicates no significant change across the AP axis. Also, the availability of the μ CT scans should allow for a more details quantitation of this effect across these axes at multiple traverse sections and the inclusion of representative micro CT scans showing this effect. This may shed light on whether this is a general change in bone shape or perhaps localized to specific regions according to mechanical strain giving insight into the underlying biology.

Finally, in line with comment 4 above and the established role of *Qsox1* as a modifier of extracellular matrix, a measure of bone material quality (such as micro-indentation) may help clarify the whether changes in bone geometry due to *Qsox1*^{-/-} are accompanied by changes in bone material composition, and thus if the changes in bone geometry may be compensatory.

Minor comments:

Table 1 - needs legend indicating the significance of bold typeface - likely established bone genes. Line 91-92 may be interpreted as though RNA-seq was performed on more than 600 mice. Should clarify.

Line 153: Unmatched close bracket.

Line 179: written 'osteoblast-mediated' instead of 'osteoblast-mediated'.

Line 184-186: Perhaps best framed as how they may impact BMD – how it is written currently overstates the strength of evidence.

Line 270: Not clear if merge analysis is a pre-existing analysis approach or something developed by the author. This should be clarified and perhaps denoted by italics.

Line 618: written 'BNs' and I believe it should say Bayesian networks as this abbreviation has not

been used previously.

Line 648: Says 'Analysis of BMD data on Rasd1-/-...' when I believe it should say 'Analysis of BMD data on Glt8d2-/-...'

Reviewer #4:

Remarks to the Author:

In their manuscript, Al-Barghouthi et al. use systems genetics tools on a rich resource they have generated in the Diversity Outbred (DO) mice to dissect bone-related traits. They first use RNAseq data to define "Bone-associated nodes" or BANs, essentially genes related to bones through WGCNA clustering, Bayesian Network construction, and gene ontology analyses. To find potentially causal BANs, they then looked at whether their human homologs are from BMD GWAS or have an eQTL in any human tissue. The authors then perform a GWAS in the DO and find little overlap with human GWAS for BMD. Finally, they characterize one QTL in chromosome 1 that influences multiple bone traits and identify *Qsox1* as the causal gene through the generation of *Qsox1* mutant mice. While the data on which the work is based will serve as a valuable resource upon publication, the manuscript and the presentation of the data require considerable improvement in order to be accepted.

Major points:

- The manuscript tries to do a lot of things but fails to do any of them with a minimum of completeness. For example, the authors directly use WGCNA to generate modules of correlated genes without any diagnostics about the RNAseq quality (PCA, MDS) and no differential expression based on sex or any other parameter.
- Regarding the network analysis (WGCNA followed by Bayesian), the authors identified 1050 candidate genes. Do they include the "known bone genes"? If so, why don't they include all known bone genes? If not, why didn't they add the known genes to the associated ones for the rest of the analysis? In general, it seems that they wanted to reduce the dimensionality of the data using complicated methods but the reduction is not quite significant (from 1539 to 1050 genes). Moreover, they started with the known genes and reached to the associated genes which its reason is not quite clear to me.
- The authors overlaid the results of the network analysis with the human QTLs and identified 46 genes where 37 of them were originated from the known list. It seems that quite a few genes from the known bone genes were not included. How would be the situation if they didn't perform any network analysis and started with the 1539 genes? They claimed 9 new candidates but they might have lost some known genes as they didn't include the 1539 genes.
- Have the authors ever considered potential confounding factors in their network analysis?
- The scRNA data that they generated is missing in the fig1. They need to add this. How many cells of how many mice they measured? In addition, the authors mention in the text that the scRNA-seq was performed on mouse bone marrow-derived stromal cells (lines 220-221) but in the Methods section they describe the protocol of osteoblast differentiation. Further clarification of this discrepancy is required.
- For every dataset generated, the authors could perform some high level / conventional omics analysis. For instance, the scRNA data were only used to check the expression of two genes across the cells.
- The order in which the results are presented lacks coherence or purpose. The authors start with RNAseq analysis (without looking at eQTLs), then they integrate with human GWAS and human eQTL data, then they perform the DO GWAS, and after the DO eQTL. Finally, they focus on experimentally

validating one locus of their choice. The manuscript may read much better if the flow is optimized. As a suggestion, one could first perform all mouse analyses, integrate with human data, and then finalize with the experimental validation.

- To make the manuscript more readable, it would be great to write down why a given analysis was performed. This is particularly important for those who have less computational background.

- In general, it feels like most of the results are in supplementary figures or tables while the main figures and results convey very little information. This is especially the case for the RNA-seq part at the beginning, but also true for the DO QTL and eQTL data. On the same line, the supplementary tables are very difficult to understand and a graphic representation of the results will significantly help the reader to understand the data.

- Are the 192 mice used for RNAseq also phenotyped with the other mice? Are they part of the 619 mice? They seem to be phenotyped based on supplementary table 8, but this should be clarified in the methods.

- The motivation behind the choice of computational strategy to identify BANs is not clear to me. Specifically, why did the authors choose to identify modules of co-expressed genes (using WGCNA signed co-expression network), followed by Bayesian Network construction on the genes of each module? There is little value (and it makes little sense) to perform Bayesian Network construction using features that are already positively correlated. Has this been done before?

- Based on the author assumption, *Qsox1* deletion should have resulted in a strong bone phenotype and this doesn't seem to be the case. Can this come from the fact that the phenotyping data obtained from different CRISPR/Cas9 lines were combined? Or is it due to the mixed genetic background used (B6SJF1xC57BL/6J – random selection of 50 F2 male mice)?

Minor points:

- It may help that the computational strategies (which are in many cases custom) be explained graphically.

- Stylistic note: it is best to avoid nested parentheses (e.g. lines 66,68), as well as whole sentences in parentheses (e.g. line 67).

- Line 155: The relevance of the following phrase is not clear to me: "indicating the structure of the Bayesian networks was not random with respect to connectivity". Specifically, why would one expect that genes affecting bone traits to be more connected?

- Line 161: was there correction for multiple testing?

- Line 626: For the hypergeometric test, you refer to "total number of bone genes in our bone gene set". Are these all expressed in your RNAseq dataset? It may be a good idea to remove those that are not expressed.

- Line 238: There is no mention of how the correlations of phenotypes (in Supplementary Table 3) could affect the QTL mapping results. This is especially relevant for the Chromosome 1 locus that the authors define as "pleiotropic" (Line 294). Is it really pleiotropic because the locus is affecting multiple traits? Or because the traits are highly correlated?

- The selection of all GTEx tissues instead of a subset is not well explained. In addition, how do the data from different tissues overlap with human BMD GWAS? In other words, are there tissues that overlap more than others?

- You may want to better differentiate the human eQTLs and mouse eQTLs in the text to avoid confusion.
- You should explain what PPH is, and why it is used.

Response to Reviewers' critiques of Al-Barghouthi et al. "Systems genetic analysis in Diversity Outbred mice informs human bone mineral density GWAS and identifies Qsox1 as a novel determinant of bone strength"

We appreciate the informative and thoughtful comments from all four reviewers. They raised excellent points and, as a result, the revised version of our manuscript is substantially improved. Below we address each of the critiques. Our responses are highlighted in bold red type following each critique. Additionally, all changes made in the manuscript have been tracked.

In addition to the responses to the reviewers' comments outlined below, we identified mistakes that we have corrected in the revised manuscript. Below is a summary of these changes.

- 1. Our initial analysis used GWAS SNPs (1,103) from an eBMD GWAS, but (erroneously) also included partial colocalization results from another GWAS, which interrogated femoral and lumbar spine BMD. We found that it was more informative to include both GWASs, as they incorporated additional information. The changes we performed essentially increased the number of GWAS SNPs from 1,103 to 1,161 (added 58 SNPs), and increased the number of colocalizing eqtl slightly. Importantly, no previously colocalizing BANs were removed, and none of our conclusions were altered. Changes are mostly contained within Table 1 and the "Using BANs to inform human BMD GWAS" section of the results.**
- 2. Table 2 contained an error in one cell, and this has been corrected.**
- 3. We have slightly changed the p-values of Qsox1 knockouts in the last section of the results, for the three caliper-based measures (ML, AP, FL). We used a type II ANOVA, instead of a type I ANOVA, which is more suitable for unbalanced designs, and returns more accurate p-values. This does not affect our conclusions.**

Reviewer #1 (Remarks to the Author):

In this manuscript, Al-Barghouthi et al. utilize DO mice to map and identify genes that play a role in a variety of bone-related traits. Integrating these studies with gene expression waastudies in mice and humans together with human GWAS data, the authors report cumulative evidence supporting the role of a number of candidate genes in mouse bone development, as well as potentially towards interpreting human GWAS data, although the link between these is less convincing. The manuscript culminates in the generation of Qsox1 knockout mice which are used to demonstrate that this gene is required for optimal bone development and function. The manuscript is well written, the methods are well detailed, and many novel insights were

discovered. It's not clear to me that all of the claims in the manuscript are strongly supported by the evidence, however it nonetheless represents an important advance in our understanding of the genetics of bone-related complex traits.

Concerns:

1. It's not clear that the statement in the abstract on line 45 and elsewhere in the paper describing the genes identified as "likely to be causal for human BMD GWAS" is fully supported by the evidence.

For example, for *Glt8d2* this claim is based on:

- a. Results from a knockout mouse that is probably unlikely to mimic the effect from a GWAS SNP/haplotype and is associated with partial lethality.
- b. An eQTL from a potentially unrelated tissue/cell type (although in this case pituitary could be relevant). Also, my understanding is that the eQTL was based on proximity to the BAN and not linked to the risk GWAS allele which means the GWAS allele may not be associated with any expression differences in *Glt8d2*.
- c. Expression within the bone marrow.

I certainly agree each of these add to the support that *Glt8d2* could be causal, but without any context of what is known about nearby genes or specific effects on expression or function based on the risk GWAS haplotype, it seems premature to claim that the causal genes have been likely identified.

Thank you for this comment. Let us try to clarify. This comment seems to focus on the lack of a direct link between BMD-associated SNPs and *GLT8D2* (as well as all the other bone associated nodes (BANs) that we identified and suggest are causal). However, there appears to be confusion as to how we used human eQTL data. In our approach we first used the network-based approach to identify BANs. To identify BANs potentially responsible for the effects of BMD GWAS loci, we filtered those BANs located within GWAS loci that were also regulated by a colocalizing eQTL in a human tissue. By colocalizing we mean statistical evidence that a BMD locus and eQTL were driven by the *same* variant providing a direct link between BMD-associated SNPs and BANs (including *GLT8D2*).

We believe that many of the 72 genes we identified using our network-based approach are truly causal due to the following reasons. First, the network analysis identified genes (BANs) with evidence of an involvement in a bone regulatory process. Second, the identification of colocalizing eQTL identified genes whose expression is influenced by BMD-associated variants. Together we have connected BMD variants to the regulation of target genes and target genes to bone regulatory processes. As we mentioned in the discussion, we believe both lines of evidence (including the additional supporting experimental evidence for *GLT8D2* and *SERTAD4*), suggest these genes are causal.

2. How many BAN genes were also on the list of known bone genes? Is the 29 known bone gene enrichment on line 175 based on the GTEX eQTLs further enriched with genes on the known bone list relative to the whole list of 900 BANs with human homologs or 544 near the GWAS SNPs?

NOTE: In response to a comment from reviewer #4, we updated the analysis which changed the number of BANs. To avoid confusion, below we use the previous values as well as provide the new values (in parentheses).

Of the 900 (1251) BAN genes with human homologs, 312 (388) were in the known bone gene list. Of the 544 (738) BANs within 1 Mbp of a GWAS SNP, 209 (257) were in the known bone gene list. The gene enrichment is the enrichment of known bone genes within the group of genes that were both BANs and had colocalizing eQTL relative to the number of known bone genes in the 544 (738) BANs within 1 Mbp of the GWAS SNPs. We have edited the text for clarification.

3. How much of the genome is covered by the 1,103 BMD genome SNPs ± their 1 MB window? Are the BANs enriched for being near GWAS SNPs?

Approximately 36% (~1.1 Gbp) of the genome (~3.1 Gbp from GRCh37.p13) is within +- 1Mb window around the 1,103 GWAS SNPs from Morris et al. [1] . As noted above, when we combined associations from Morris et al. and Estrada et al. [2] for a total of 1,161 SNPs. This increases to ~1.11 Gbp, returning essentially the same coverage of the genome.

To answer your second question, we overlapped all protein coding genes in the genome (22812) with the genes overlapping the GWAS loci windows (+- 1 Mb;10810 overlapping genes). We then calculated enrichment of the BANs being located within these windows, relative to the protein coding genes in the genome being near GWAS SNPs, using the hypergeometric distribution in R:

NOTE: This calculation was performed using the new number of BANs

**q=738, k=1251, m=10810, n=22812-10810
phyper(q-1,m,n,k,lower.tail=F).**

The p-value is 1.7×10^{-17} , indicating that BANs are highly enriched within GWAS loci. We have added this information in the results section (pg. 9, line 273)

4. It would be nice if Table 1 included the tissue(s) that the eQTLs were based on.

Great idea. We have added a column with the tissue from which that colocalization probability was calculated (tissue of strongest colocalizing eQTL) to Table 1.

Furthermore, we have added a supplementary table (S9) with all significant (PPH4 \geq 0.75) GWAS-eQTL colocalizations and their relevant tissues.

5. The IMPC data for the *Glt8d2* KO is used to identify an increase in BMD in male mice (Fig. 2e). Can the authors comment on why the IMPC website lists no significant results for these mice for any skeletal trait including cortical BMD in the KO mice? Same question for *Qsox1* which is also in the IMPC database and has no significant skeletal phenotypes.

In the case of *Glt8d2*, the IMPC did not consider weight as a covariate in its models. Given the strong relationship between body weight and whole body BMD, we used the Phenstat R package [3] (designed for the analysis of IMPC data) to correct BMD for the effects of weight as described in the paper. After adjusting for body weight the effect of *Glt8d2* genotype had a significant effect on BMD as reported in the paper. In their most recent update, the IMPC overhauled their analytical approach to include weight as a covariate. As can be seen on their website, the BMD data for *Glt8d2* homozygous knockouts has a p-value of 1.97×10^{-3} (can be accessed here: https://www.mousephenotype.org/data/charts?accession=MGI:1922032&allele_accession_id=MGI:4364018&pipeline_stable_id=MGP_001&procedure_stable_id=IMPC_DXA_001¶meter_stable_id=IMPC_DXA_004_001&zygosity=homozygote&phenotyping_center=WTSI).

In the DO, the Chromosome 1 QTL, for which *Qsox1* is supported as causal, specifically influenced phenotypes related to femoral size and cortical bone accrual along the medial-lateral axis of the femur. As described in the paper, these exact phenotypes were altered in mice deficient in *Qsox1*. The IMPC used dual-energy x-ray absorptiometry (DEXA) to measure BMD of the entire skeleton minus the skull. It is unlikely these very specific femoral phenotypes would manifest as a change in whole body BMD.

6. The manuscript mentions another cohort of DO mice (line 298). Has this data been published elsewhere? If not, a fuller description of this cohort and its QTL analysis would help support its use as replication of the Chr 1 loci, as well as including whether or not this cohort supported any of the other QTL loci.

Great point. Yes, this cohort has been published [4]. We cite this work in the methods under “Prior ML QTL mapping” as well as provide a brief summary of the cohort and how we performed the QTL mapping for medial-lateral femoral width, and we have added a citation to the results section as well. We only measured caliper-based phenotypes in this cohort, and only the Chr 1 ML QTL was replicated. We have added a statement in the results section to provide more details (pg. 19, lines 499-502).

Typos:

1. Line 85. Should say “heterogenous” instead of “heterozygous”??

We meant to say heterozygous and we believe this is correct. Our intent was to distinguish the high heterozygosity found in the DO to other widely-used mouse genetic reference populations comprised of inbred strains.

2. Line 167. Should say “causal” instead of “casual”.

We have corrected this oversight.

3. Line 179. Should say “mediated” instead of “medicated”.

We have corrected this oversight.

4. Line 286. “identify” instead of “identifying”

This was meant to mean “we calculated [...] by randomly selecting [...] and identifying. We have edited the text for clarification (pg. 18, lines 452-454).

Reviewer #2 (Remarks to the Author):

Al-Barghouthi et al submitted a paper evaluating bone phenotyping data obtained from Diversity Outbred (DO) mice to find new candidate genes associated with bone-disease-related traits. The authors point out that there are many associations known for links between genetic variants and osteoporotic traits. However, these associations are mainly linked to changes in bone mineral density only, not making use of other traits that could be relevant as a biomarker for bone disease. They applied a network approach to cortical bone RNA-seq data and found 46 genes with a presumably causal link to human bone disease some of them novel in the field. Candidates were validated using existing or newly generated phenotyping data. In the next step, the authors performed a GWAS in their DO cohort using a wide range of bone traits and identified Qsox1 as new candidate gene affecting bone structural features.

The analysis addresses the very relevant disease area of osteoporosis affecting not only patients in the U.S but all over the world impacting health and quality of life, especially in aged patients. Several analyses have been published already attempting to identify candidate genes associated with human bone disease based on data generated from experimental models such as laboratory mice. These candidate genes and the genetic networks respectively physiological pathways are thought to offer new targets for the prevention and treatment of human disease.

The presented analysis comprises an impressive range of methods both regarding bone-related phenotypes in DO mice and additional mouse models used to validate the findings as well as network and genetic analysis to identify new candidate genes associated with bone disease and gain additional evidence for a link to bone metabolism.

Overall, the authors present a solid and interesting genetic analysis resulting in a small number of new candidate genes. The role of these candidate genes in bone metabolism are discussed. I have only a few questions/remarks that need clarification from my point of view:

It was not quite clear to me, which genes dropped out as candidates during the different analysis steps (also see remarks below). The author state that there are over 1000 known candidates so I also wonder why only a few new candidates were detected.

Sorry for the confusion. Our statement in the introduction was that GWAS has identified over 1000 BMD associations. For nearly all of these associations, we have no idea which genes are responsible. The purpose of our network analysis was to identify candidates for a subset of these loci. We did not expect to identify candidates for all loci.

As for the “few number of new candidates” that were detected, there are a few things to keep in mind. First, our approach was purposely stringent. We required that a candidate must be proximal to a GWAS locus, must be significantly connected with more bone genes than expected by chance in the network analysis (supporting its role in the regulation of bone), and must colocalize with a human eQTL.

Second, keep in mind that our colocalization analyses using GTEx eQTL was limited, as there were no eQTL data from bone tissues/cells. While this does not diminish the validity of our results (due to many eQTL being shared across tissues [5,6] and effects from non-bone tissues impacting BMD), we are likely missing a number of key colocalizing eQTL specific to bone cells.

As for gene dropouts due to quality control, pruning, and so on, we have amended the methods to include the numbers of genes that dropped out after each pruning step (as mentioned in “minor remarks” below).

While reading the introduction and results sections I was wondering what are the founders of DO and do they show differences in bone-related parameters. Later I realized that this information is provided e.g. in Suppl. Figure 1 – could these details be presented more prominently?

We have added the information on the DO founders to the introduction. We did not phenotype the DO founders in this study. To our knowledge the complete set of phenotypes we measured in the DO have not been measured in the DO founders. In a recent collaboration we did measure some of the same phenotypes as part of a study to evaluate the effects of unloading on bone [7].

The authors point out that so far analyses are mainly based on BMD data not making use of a more in-depth analysis of structural characteristics of bone presumably related to bone disease. Here, I wonder 1) what is the translational potential for changes detected in mice and the link to human clinics, and 2) are the 50 complex traits somehow inter-correlated, or do they occur in

syndrome-like clusters? Were the traits treated as independent in this analysis? Could there be a kind of “hierarchy” for these traits?

To address your first question, we believe that the mouse is a powerful tool to unravel the molecular basis of bone strength. We demonstrate this in the first part of the manuscript in which we identify genes likely causal for human BMD GWAS loci by performing a network analysis of mouse bone RNA-seq data. In the second part of the paper, we focus on GWAS for traits that are clinically important, but not amenable to human GWAS. Through this analysis we identify *Qsox1* as a regulator of cortical bone accrual and strength. Human *QSOX1* had not been implicated by human BMD GWAS, demonstrating our ability to highlight novel genes using the DO that are involved in the regulation of a clinically important phenotype. In future studies we plan to directly evaluate the clinical potential of targeting *QSOX1* to increase bone strength.

It should be noted that we also identified genomic regions, syntenic in mice and humans, in which both harbored GWAS associations for BMD in humans and strength-related traits in the DO. Although it is outside of the scope of the current work, in future studies we plan to focus on these regions. It is likely that the more detailed phenotypes collected in the DO will provide novel insight into the mechanistic basis of how these overlapping human loci impact BMD and this, in turn, could have clinical implications.

With regards to your second question. Yes, the 55 traits are measuring different aspects of bone and all are related to bone strength. That said we do see correlations between traits, and in some cases the correlations are quite strong. This is especially true of traits within the same category (e.g., microCT traits). In the GWAS, we treated these as independent and these correlations were reflected in the findings. For example, the locus on Chr. 1 affected traits related to the cross-sectional size of bone (Medial-lateral femoral width, total area, etc.) - all traits that were highly correlated. Although we feel it is outside of the scope of the current paper, in future analyses, it would be prudent for us to focus more on the relationships between traits to truly understand how identified QTL are truly impacting bone.

There are a couple of minor remarks that may require clarification:

Line – 82 – please list the founder strains here

We have added this information (pg. 4, lines 87-88).

Line 111 – here it is stated that 55 complex traits were measured – not 50?

We measured 55 traits, perhaps the confusion is due to us using “over 50” elsewhere in the text. We have changed these to “55”.

Line 126 – were these mice food-deprived for 24 hours? Why so long and could that affect bone marrow adiposity or other parameters?

Thanks for pointing this out as this was a mistake in the original version of the manuscript. The mice were actually fasted overnight (16-18 hours). This has been changed in the methods (pg 30, line 783). Overnight fasting is standard procedure in mouse studies. Our main reason for fasting is that although not reported in the paper, we measured a number of metabolic phenotypes including blood glucose at sacrifice, and fasted glucose measurements are the gold standard. Although we have not tested this, it is also possible that fasting would reduce variation in diet- and feeding-response gene expression changes in bone.

To our knowledge, the effects of overnight fasting on bone in mice has not been tested; however, in rats during a 4 day fast vertebral bone density and morphology did not change after 24 hours of fasting [8]. Therefore, we feel it is unlikely an overnight fast would have an appreciable effect on bone morphology, microarchitecture, biomechanics, or histomorphometry in the DO. In the context of marrow adiposity, it has been shown in humans that marrow adiposity does not change after a short fast period [9]. Additionally, it has been shown in rabbits that acute starvation does not impact marrow adiposity [10].

Line 132 – what exactly does 30.8-fold “variation” mean?

In the BV/TV data, the highest measurement was 30.8 times greater than the lowest measurement. We have added a clarification in the text (pg. 7, lines 173-174).

Line 229 – IMPC data was used for validation – did this mutant line have further phenotypes (e.g. metabolism or X-ray)?

Yes, the IMPC measures a large battery of phenotypes in mice. According to the IMPC, phenotypes significantly affected in *Gltd2* knockouts were “decreased total body fat amount”, “abnormal body weight”, and “preweaning lethality, incomplete penetrance”.

Line 264 ff – this statement is somehow difficult to understand and appears a bit speculative.

We have removed this sentence.

Line 286 – to identify instead of to “identifying”?

This has been corrected.

Line 294 – do the authors interpret effects on several complex traits as “pleiotropic”? Is this justified or is it an effect on the same feature just affecting several complex traits?

Great point - we have removed the use of “pleiotropic”. In this case you are correct, its likely the locus is affecting the same aspect of cross-sectional size that is reflected in multiple traits capturing slightly different aspects of cross-sectional size.

Line 493 – please specify the “standard chow” diet – manufacturer, product number, chow, or purified diets? These details should always be provided to allow the exact replication of the results.

We have added this information to this section, and to other relevant sections in the Methods.

Line 494 – see above: why were mice food-deprived for 24 hours? May that affect any of the parameters e.g. bone marrow adiposity?

See answer our response (6th response to reviewer #2) above.

Line 495 – “approximately 12 weeks” – what was the range? Why did the authors choose relatively young mice? I assume age affects bone phenotypes.

The median was 86 days, with a range of 76-94 days. This has been added to the Methods. We chose this age because strength will reflect the actions of bone accrual and will not be confounded by bone loss, which occurs in most inbred strains by ~16 weeks [11].

Line 565 ff – In this section, a couple of exclusion criteria were described. It would be good to know how many genes were excluded because they failed QC.

We have added this information.

Line 583 and 589 – how many genes were removed from the analysis due to QC issues?

We have added this information. However, it is worthwhile noting that in the mouse genotyping section, genotyping markers were removed, and not genes.

Line 655 – why “both mutants”? – not clear

We have removed “for both mutants”.

Line 671 – why was “DO mouse generation” included as a factor – did mouse generation vary a lot?

The DO generation ranged from G23 to G33. DO generation is controlled by the Jackson Laboratory’s breeding scheme. The lab produces and distributes 4-5 generations of mice per year. We tested the effect of generation and it was significantly associated with

nearly all phenotypes. For example, see the figure below, which shows the effect of generation on bone strength (max load) in our cohort. To remove this variation, we used generation as a covariate. This is a common source of variation that is often controlled for in DO QTL mapping studies (e.g., see Yuan et al. 2018. *Mammalian Genome* [12]).

Line 715 – specify “standard chow diet” (see above)

We have added this information.

Line 717 – specify the HFC, low-fat, and high protein diet (see above)

We have added this information.

Line 718 – age here 24-25 weeks – why were mice older?

These data were from a separate study that we used to replicate the Chr. 1 femoral size QTL. The study utilized different endpoints relevant to that study [4]. A description of this population is provided in the methods section (“Prior ML QTL mapping”).

Line 772 – for Qsox1 mutant mice a C57BL6J was used – Rasd1 was likely on an N background – why did you change the background? Could that have a consequence on bone phenotypes?

To answer this question, first we would like to mention that the gene name “*Rasd1*” was erroneously used in place of “*Glt8d2*” in a section header in the methods and this has been fixed.

We do not anticipate the background to have a consequence on our analysis in this study. In the generation of *Qsox1* mutant mice we used C57BL/6J due to the use of this strain by our gene targeting core. However, we are not comparing *Qsox1* CRISPR/Cas9-mediated knockout line with any other mutant mouse lines. We are only comparing the effects of *Qsox1* genotype (wild-type vs. null) within the C57BL/6J background.

In addition:

1000 loci associated with bone phenotypes can be found in the literature - can this list be provided as supplemental information?

The “over 1000” loci are the results of many GWAS studies performed on many related bone phenotypes (mainly BMD). As such, it is difficult to include them in a list. However, the data used for this study are available at (<http://www.gefos.org/?q=content/data-release-2018>) and (www.gefos.org/?q=content/data-release-2012), and these links can be found in the “data availability” section of this paper. Furthermore, as the list of loci is always changing, we recommend visiting the GWAS Catalog (<https://www.ebi.ac.uk/gwas/>) for a frequently updated list of GWAS loci.

Reviewer #3 (Remarks to the Author):

Reviewer summary: In this study, Al-Barghouthi et al. measured >50 skeletal traits in a cohort (N=619) taken from the Diversity Outbred (DO) mouse population and use this to discover molecular features correlated with changes in bone morphology or function. Generating immense transcriptomic and genotype datasets, their multimodal data analysis approach identifies genes that influence skeletal phenotype in mice and potentially contribute to human skeletal trait variance.

In the transcriptome analysis, key genes were identified as those co-expressed in mouse bone tissue and strong network connectivity with genes previously shown to play a role in bone. The relevance of these genes to human bone mineral density (BMD) variation was inferred based on 1) the colocalization of human orthologs within 1Mb of a significant BMD-GWAS loci and 2) the cooccurrence of eQTLs which regulated these genes in the same region. Among those, *Sertad4* and *Glt8d2* were specifically examined as putative regulators of BMD, with the latter shown to impact BMD in mice. Hypotheses regarding the mechanism by which these genes impact skeletal biology were then developed, integrating network information, bulk and single-cell level transcriptome data to identify plausible cell-types and pathways.

Using the genotype data, QTLs associated with skeletal traits were identified by GWAS in mice. The overlap of these syntenic regions with BMD-associated regions in the human genome was determined. As this overlap was not statistically significant, the authors took this to indicate that

the mouse QTLs were driven by genes involved in processes influencing aspects of bone other than BMD. To identify potential causal genes associated with the QTLs, genes containing deleterious missense variants significantly associated with trait variation and a colocalizing eQTL were identified. This identified Qsox1 as a strong causal candidate, which the authors went on to show impacts bone geometry and strength when functionally deleted in mice.

The article is creative, well written and provides possibly the largest data resource of its kind in the field of skeletal biology. As an approach, this article demonstrates how molecular genetic data from mice can complement and enrich data from human populations, guiding the search for causal genes underlying association signals and provide insight into important skeletal characteristics not measurable in humans at scale. While I am confident the data, analytical approaches and causal gene candidates will help guide future investigations into skeletal genetics, there are areas of the manuscript that should be clarified or strengthened with additional analyses and data. I have outlined these areas in the comments below.

Comments:

1) The utility of the data resources hinges on accurate meta-information, including sample relationships between datasets, inter-trait relationships and feature/gene information. There are several errors in the first supplementary figures and tables describing the phenotypes. For example:

- Supplemental figure 1 - panels b and c do not correspond with the figure legend. Panel c - it is not clear how the traits are grouped/colored, it would appear to be based on the phenotype category. Needs this information in the figure legend.

Yes, traits are colored by phenotypic category. We have added this to the figure legend. Also, the panel labels were fixed in order to correspond with the legend.

- Table S1 - abbreviations duplicated for 40 of the phenotypes (all histo_OV/TV). Also, not immediately clear how 'Number measured' and 'Number mapped' are defined. More detail required in the Table legend.

The abbreviations seem to have been duplicated during upload. We have fixed this issue, and added the following description to clearly define “number measured” and “number mapped” to the table legend.

Phenotypes collected in the DO. Includes phenotype, abbreviation, measurement units, number measured and number mapped. “Number measured” corresponds to the number of measurements available per phenotype, while “number mapped” indicates the number of measurements included in the QTL mapping. Since QTL mapping utilizes covariates, mice with missing covariates (such as weight) were excluded from mapping.

- Using gene symbols as feature identifiers (such as gene lists etc.) can be troublesome as gene names change over time with new information and commonly used software (i.e. excel)

can misinterpret these character strings leading to mislabeling. It is recommended that stable, unique gene identifiers, such as Ensembl ids, be included in tables along with gene names to ensure the temporal fidelity of results across platforms. Currently table S4, S6, S7, S11 and S12 only have gene symbols as identifiers.

We have now added stable, unique gene identifiers (MGI IDs) to the listed supplementary tables.

- Line 129-131 and Table S2-3: The p-values are very low for traits that have a relatively weak correlation with bone strength ($|r| < 0.3$, table S2). This discordance may be due to the use of Pearson correlation which relies on a number of assumptions regarding the underlying data distribution which may not be valid across such a diverse spectrum of measured traits. Spearman correlation as an alternative requires less assumptions and may be more suited in this context. This should provide a better estimate of the relationship between traits.

We have amended S2 and S3 (now S3 and S4) to include Spearman, instead of Pearson, correlations. Furthermore, we realized some mistakes in these tables which have now been corrected. Also, in the original S3, the upper diagonal contained p-values that were adjusted for multiple testing, while the bottom diagonal contained unadjusted p-values. We have amended this to only include the adjusted p-values.

However, generally, there is little appreciable difference in the correlations and p-values between the Pearson and Spearman methods. Instead, it is likely due to our being well powered to detect “weak” correlations due to our sample size (N=up to 619).

- One of the key strengths of this data resources is having matched phenotypic, genotype and transcriptomic data on such a large cohort. To ensure this aspect of the data can be utilized by the wider research community, it is recommended the authors include an additional table containing raw phenotyping data for each mouse, and another table to link this phenotyping information to genotype and transcriptomic data between samples. Additionally, the authors should provide more information as to the design of the RNAseq dataset e.g. How were the samples for transcriptomic analysis (N=192) chosen from the entire cohort (N=619)? It should also be explicitly state how many mice were genotyped in figure 1, results or methods.

We have added a new supplementary table (S2) which includes data on all 55 phenotypes for all mice, and all covariates required for mapping. Furthermore, all of our data is linked by Mouse ID (sample name). We have added raw genotyping data, as well as calculated genotype probabilities, allele probabilities and other relevant files to Zenodo, a public data repository (DOI:10.5281/zenodo.4265417 - see “data availability” section). All transcriptomic data has been deposited on GEO (see “data availability” section) and all data are linked via Mouse ID. Also, mapped QTL/eQTL and the raw data can be accessed and downloaded from our website (<https://qtlviewer.uvadcos.io/>).

Regarding the design of the RNA-seq experiment, the 192 samples were randomly chosen from the available mice at the time (mice 1-417), with the constraint of having an equal number of male and female mice. This has been clarified in the methods section, under “Bulk RNA isolation, sequencing and quantification” and in the first section of the results.

Furthermore, all 619 mice were genotyped. This has been clarified in the methods section under “Mouse genotyping”, Figure 1, and in the first section of the results.

2) Regarding the identification of 'bone associated nodes' (BANs) that have a potentially causal role in human BMD variation described in section 'Using BANs to inform human BMD GWAS'. It would be helpful if the authors could elaborate on the motivation for using both proximity to GWAS signals in syntenic regions and the colocalization of a local eQTL to prioritize BANs that are more likely to be causal. While the return of such a high fraction of genes with established role in bone seems to support such a strategy, some clarification around the hypothesis driving such a filtering approach may be warranted. i.e. Are genes that are located in proximity to a given association signal more likely to be 'causal' if they colocalize with an eQTL?

Our approach to identifying genes likely to be causal for human BMD GWAS associations relied on two key pieces of information; 1) genes being identified as BANs in Bayesian networks generated from DO cortical bone and 2) BANs being regulated by a colocalizing human eQTL. We reasoned that together both pieces of information provided strong support for genes as being causal for GWAS loci. The rationale being that the identification of a gene as a BAN provides support for that gene being involved in a bone regulatory process, and the identification of a gene having a colocalizing eQTL provides a direct link between BMD-associated variants and the expression of a gene. With these data we have linked BMD-associated variants with the molecular perturbation of target genes and linked target genes with a bone regulatory process. We have added more information in the results section (pg. 9, line 275-284) to clarify our rationale.

As for whether GWAS proximal genes (+/- 1 Mbp from a GWAS SNP) are more likely to be causal when they colocalize, we believe so. If we were to take all genes proximal to the BMD GWAS SNPs, this would result in ~10,000 genes, roughly half of all characterized protein-coding genes. Additionally, a common practice in post-GWAS analyses is to take the closest gene, however studies have shown that the closest gene is often (and more often than not) not causal [13]. Studies have also shown that proximity of genes to GWAS SNPs is informative for identifying causal genes[14]. Finally, studies have also shown that both proximity of a gene to a GWAS SNP and colocalization with eQTL each provide an additional source of causal evidence when conditioning on the other[15]. Therefore, in order to highlight genes that are more likely to be causal, we identify genes that are proximal to GWAS SNPs, are regulated by a colocalizing eQTL, and show evidence of being involved in a bone regulatory process.

3) In the identification of potentially causal genes associated with the trait GWAS signals in mice (Section beginning line 268), the authors impute variant information based on the GigaMUGA array and individual strain genomes, before selecting missense, trait-associated variants. As an alternative/supplementary approach, I would encourage the authors to call variants using the transcriptome sequencing data as this may be a more direct approach to mapping variants in individual mice. Further, by comparing the results of the imputation and with RNAseq variant calling across the 192 mice in the transcriptome cohort, the authors could establish the fidelity of the imputation approach and whether it is appropriate for use on the genotyped cohort (>600 mice). While this will somewhat bias the resolution to exonic regions, as the current prioritization approach selects only missense variants this should not be an issue. Further it may allow the direct comparison of phenotypes between mice with and without variants of interest, such as those with or without Qsox1 variants.

This is a good point. Just to clarify, we believe that the essence of your question is, could we do a better job of identifying missense variants potentially responsible for QTL effects by calling variants using our RNA-seq data. Here is why we do not think this is the case. Our approach started by identifying missense variants that were among the variants most significantly (top 15%) associated with a given bone trait in our DO. The missense variants used in the analysis were identified from the deep whole genome sequencing of the eight DO founders [16]. As such, the WGS gives us what should be a pretty complete picture of all missense variants segregating in the DO. To “call” the variants we used GigaMUGA genotypes along with algorithms developed by the Churchill lab [17] to reconstruct the haplotype structure of each DO mouse. For each QTL region, the WGS data from the eight founders were “imputed” onto the reconstructed haplotypes. Due to the ability to reconstruct the haplotypes very accurately (due to the fact that each haplotype in the DO can be clearly assigned to its founders of origin), the accuracy of imputing the comprehensive set of variants from WGS is high.

Choi et al. has recently developed a “genotyping by RNA-seq” (GBRS) algorithm which performs the haplotype reconstruction step using RNA-seq data instead of GigaMUGA genotypes [18]. The program does this by calling known variants (from the WGS study) from RNA-seq data to reconstruct founder haplotypes. It has been shown to have similar performance compared with haplotype reconstruction using GigaMUGA genotypes [18]. The advantage of GBRS is not necessarily increased accuracy of haplotype reconstruction, but rather a reduction in cost by not having to perform genotyping by array for samples where you have RNA-seq data. The obvious limitation is that to perform haplotype reconstruction using GBRS requires RNA-seq data on all samples, which is not the case for us (we have RNA-seq data on 192 of the 619 mice included in the study). As a result, it is unlikely that any additional information could be gained regarding missense variants present in the founders and segregating in the DO from using RNA-seq data.

The one case where calling variants from RNA-seq data may be informative would be the case of de novo variants arising within the populations. While such variants certainly

arise in the population, they are likely rare and would be much less likely to contribute to phenotypic variation compared to variation from the DO founders. In addition, it would be challenging to use RNA-seq to identify new variation. First, while it is certainly possible, there are no developed algorithms designed to do this in the DO. GBRs does not call novel variants. Calling variants from RNA-seq is an active area of investigation. GATK has been developed for work in human samples. However, it is not clear if it is suitable given the nuances of the DO.

4) The authors state that one might expect the genetic effects on bone geometry and tissue mineral density (TMD) to be distinct (Lines 303-304). Yet, this would seem to contrast a long-standing idea that bone strength is determined by both its material composition and morphology (PMID: 12584605, PMID: 16723616). This may also be discordant with the result in table S3 which shows a strong correlation (>0.8) between Medial-lateral femoral width (ML) and TMD.

We agree that bone strength is influenced by both material composition and morphology. Additionally, we agree that this statement was somewhat misleading and we have modified it (pg. 20-21, line 519-527).

We should note that as mentioned above the correlations presented in the original supplemental table of correlations were incorrect. This has been corrected in the revised version. The Spearman correlation between ML and TMD is actually $s=-0.16$ ($P=0.056$).

While I agree the data in figure 3b and 3c do seem to indicate there are two distinct signals at this QTL, I offer a slightly different interpretation of these results. While the retention of TMD signal in Figure 3b indeed indicates TMD is independent of ML, the ablation of the ML signal when the model includes a term for TMD in Figure 3c may indicate this second locus affects both ML and TMD. It would help if the authors clarified their interpretation of this result and how their data may contrast or extend current understanding of the relationships between bone geometry and tissue mineral density.

This is a very good point. We agree with your alternative interpretation and have altered the text (pg. 21, lines 534-536) to reflect this.

5) Concerning the characterization of the Qsox1 functional knockout mice. Figure 5m shows a hypothesized model of the increased cortical bone accrual specifically along the ML axis. While Figure 5c shows the ML increase, to support this model it is important to show the data that indicates no significant change across the AP axis. Also, the availability of the μ CT scans should allow for a more details quantitation of this effect across these axes at multiple traverse sections and the inclusion of representative micro CT scans showing this effect. This may shed light on whether this is a general change in bone shape or perhaps localized to specific regions according to mechanical strain giving insight into the underlying biology.

Note: This figure is now Figure 6

This is a great point and we have added the AP plot to Figure 6.

The μ CT scans of *Qsox1* wild-type and mutant mice were only performed at the femoral midshaft - identical to what was done in the DO cohort. Since those same bones were used for biomechanical testing, we do not have bones that can be rescanned. We have included representative midshaft μ CT scans to demonstrate the differences in bone shape in Figure 6.

Due to the COVID-19 pandemic, we had to drastically reduce the *Qsox1* mouse colony (we had to cryopreserve most lines) and it would take several months to generate additional bones. We are very interested in understanding more regarding the precise effects of *Qsox1* deficiency; however, we feel that the detailed analysis of *Qsox1* function (which we plan to do once our lab is back up to full productivity post-pandemic) would take many months (> 1 year) and would add to, but not change the interpretations made in the current paper.

Finally, in line with comment 4 above and the established role of *Qsox1* as a modifier of extracellular matrix, a measure of bone material quality (such as micro-indentation) may help clarify the whether changes in bone geometry due to *Qsox1*^{-/-} are accompanied by changes in bone material composition, and thus if the changes in bone geometry may be compensatory.

As stated in the prior response, we currently do not have bones that could be used for more detailed analyses. Generating additional bones would take several months (> 1 year) and would add to, but not change the interpretations made in the current paper. Once our lab is up to pre-COVID productivity and we have fully restored the *Qsox1* mouse lines we plan to investigate the specific mechanisms used by *Qsox1* to influence cortical bone accrual along the ML axis and consequences of its deletion on aspects of bone such as bone quality and extracellular matrix composition and integrity.

Minor comments:

Table 1 - needs legend indicating the significance of bold typeface - likely established bone genes.

We have added a legend to table 1 indicating that known bone genes are in bold.

Line 91-92 may be interpreted as though RNA-seq was performed on more than 600 mice. Should clarify.

We have made this correction.

Line 153: Unmatched close bracket.

We have made this correction.

Line 179: written 'osteoblast-medicated' instead of 'osteoblast-mediated'.

We have made this correction.

Line 184-186: Perhaps best framed as how they may impact BMD – how it is written currently overstates the strength of evidence.

We have made this correction.

Line 270: Not clear if merge analysis is a pre-existing analysis approach or something developed by the author. This should be clarified and perhaps denoted by italics.

This is an established analysis and part of the Rqt1/2 R package. We have clarified this in the methods section and in the results, and have cited it on (pg 18, Lines 458-459).

Line 618: written 'BNs' and I believe it should say Bayesian networks as this abbreviation has not been used previously.

We have made this correction.

Line 648: Says 'Analysis of BMD data on Rasd1-/-...' when I believe it should say 'Analysis of BMD data on Glt8d2-/-...'

We have made this correction.

Reviewer #4 (Remarks to the Author):

In their manuscript, Al-Barghouthi et al. use systems genetics tools on a rich resource they have generated in the Diversity Outbred (DO) mice to dissect bone-related traits. They first use RNAseq data to define “Bone-associated nodes” or BANs, essentially genes related to bones through WGCNA clustering, Bayesian Network construction, and gene ontology analyses. To find potentially causal BANs, they then looked at whether their human homologs are from BMD GWAS or have an eQTL in any human tissue. The authors then perform a GWAS in the DO and find little overlap with human GWAS for BMD. Finally, they characterize one QTL in chromosome 1 that influences multiple bone traits and identify Qsox1 as the causal gene through the generation of Qsox1 mutant mice. While the data on which the work is based will serve as a valuable resource upon publication, the manuscript and the presentation of the data require considerable improvement in order to be accepted.

Major points:

- The manuscript tries to do a lot of things but fails to do any of them with a minimum of completeness. For example, the authors directly use WGCNA to generate modules of correlated genes without any diagnostics about the RNAseq quality (PCA, MDS) and no differential expression based on sex or any other parameter.

Thank you for this comment. We performed a number of different diagnostics/QC on these data, as outlined in the Methods section. We used FASTQC to evaluate sequence quality histograms, per sequence quality scores, per base sequence content, per sequence GC content, per base N content, sequence length distributions, sequence duplication levels, overrepresented sequences and adapter content. After alignment, we calculated alignment statistics to make sure all were within an acceptable range. After these QC steps, we performed PCA analyses of the expression data to identify sources of variation. The biggest sources of variation were then accounted for prior to WGCNA network construction, by batch correction using sex and age as covariates.

Furthermore, we used PEER to remove hidden confounders from the expression data for eQTL mapping analyses. This was followed with another set of PCA analyses in order to verify that PEER factors we used accounted for much of the variation.

Following that, we performed the eQTL analysis with and without correction for PEER factors to confirm that their inclusion increased power for eQTL detection. In addition to PEER covariates, we also evaluated various other covariates in eQTL mapping models to determine the most appropriate model to use for the analysis.

Our focus for this paper was on using the RNA-seq data for network generation and eQTL detection to inform human GWAS and the GWAS of bone strength traits in the DO. We did that using standard analytical procedures with appropriate QC measures. We do expect in future studies using these data we can address questions using analyses such as differential expression, but we did not feel that these would fit within the objectives of the current paper.

- Regarding the network analysis (WGCNA followed by Bayesian), the authors identified 1050 candidate genes. Do they include the “known bone genes”? If so, why don't they include all known bone genes? If not, why didn't they add the known genes to the associated ones for the rest of the analysis? In general, it seems that they wanted to reduce the dimensionality of the data using complicated methods but the reduction is not quite significant (from 1539 to 1050 genes). Moreover, they started with the known genes and reached to the associated genes which its reason is not quite clear to me.

Note: Please note that these values have changed in the revised manuscript, in response to your suggestion below, regarding the “known bone genes” list. We have kept the old values in this response for clarity.

The known bone gene list was generated by aggregating data from multiple publicly available datasets, in order to create an *a priori* list of genes that are known to affect bone from the literature. This list contained 1,539 genes. It is important to note that many of the “known bone genes” are not located in GWAS loci. We used this list to perform a “key driver analysis” [19–22]. In a key driver analysis, a “known gene” list is used to identify genes (both known and novel) that are connected in the Bayesian network

(constructed from cortical bone RNA-seq data) to more “known genes” than would be expected by chance. This led to the identification of 1050 bone associated nodes (BANs). Some of the BANs were in the known bone list, but many were novel.

In essence, we used the “known bone gene” list to identify genes (both known and novel) that were possibly involved in bone related biological processes.

- The authors overlaid the results of the network analysis with the human QTLs and identified 46 genes where 37 of them were originated from the known list. It seems that quite a few genes from the known bone genes were not included. How would be the situation if they didn't perform any network analysis and started with the 1539 genes? They claimed 9 new candidates but they might have lost some known genes as they didn't include the 1539 genes.

Note: Please note that these values have changed in the revised manuscript, in response to your suggestion below, regarding the “known bone genes” list. We have kept the old values in this response for clarity.

Of the BANs identified with colocalizing eQTL (the 47 in this case), 37 were found *in the literature* to affect bone, as defined by our “known bone gene” list. The 37 did not “originate” from this list, they were simply found in it *after* the analysis. If we didn't perform a network analysis and only used the 1539 known bone genes, then the results of our downstream analyses would identify zero novel bone genes, as all of them are already known as bone genes.

- Have the authors ever considered potential confounding factors in their network analysis?

Yes. To address potential confounders, we performed the following:

-Prior to WGCNA network construction, we pruned lowly expressed genes, as lowly expressed genes can confound analyses by introducing noise and reflecting correlations that are not meaningful.

- RNA-seq count data are heteroskedastic, meaning that variance grows with the mean. This can confound downstream analyses (such as network construction), as the results will be dominated by highly expressed, highly variable genes. To address this, we performed a Variance-Stabilizing Transformation (VST).

-We performed batch correction prior to WGCNA network construction, to remove possible confounding effects. Furthermore, expression was corrected for sex (in the full networks) and age, to mitigate possible confounding effects.

-In the Bayesian network analysis, the expression data were treated in the same manner as the WGCNA networks.

All of these points are documented in the Methods section.

- The scRNA data that they generated is missing in the fig1. They need to add this. How many cells of how many mice they measured? In addition, the authors mention in the text that the scRNA-seq was performed on mouse bone marrow-derived stromal cells (lines 220-221) but in

the Methods section they describe the protocol of osteoblast differentiation. Further clarification of this discrepancy is required.

We have added the number of mice and cells studied using scRNA-seq to Results (pg.15-15, lines 398-399) and Figure 1. We have also added detail regarding the mice and number of cells sequenced to the methods in the section entitled, “single cell RNA-seq of bone marrow stromal cells”.

In this study, we isolated bone marrow-derived stromal cells and subjected the cells to an osteoblast differentiation protocol. As these cultures are still quite heterogeneous including non-osteoblast cells, such as mesenchymal progenitors, adipocytes, and immune cells. Because of this, we felt it was more accurate to represent the cells as “bone marrow-derived stromal cells”, as they are not all osteoblasts.

- For every dataset generated, the authors could perform some high level / conventional omics analysis. For instance, the scRNA data were only used to check the expression of two genes across the cells.

This is a good point, however, a high level/conventional -omics analysis of the scRNA-seq data would have been outside of the scope of the manuscript. We simply used these data to provide support for our hypothesis that *GLTD82* and *SERTAD4* are regulators of BMD. That said, we did perform extensive QC of the scRNA-seq. For example, we preprocessed the data and confirmed its quality, identified the most variable features in the dataset to identify cell-type of clusters, verified the sex of the samples using scRNA-seq data, scored cells by cell cycle genes and regressed out those scores in order to harmonize the data, performed a clustering analysis to identify the number of relevant principal components for use in downstream analyses, generated clustering trees to identify the correct resolution for our UMAP visualizations, identified clusters which were overexpressed with our relevant genes, identified the most expressed genes within those clusters, manually annotated each cluster to identify the dominant cell type within it, and performed a Gene Ontology analysis for each cluster. While these analyses are fairly comprehensive, we only show the UMAPs with the cluster annotation, as well as the expression of three relevant genes (*Sertad4*, *Glt8d2* and *Qsox1*).

- The order in which the results are presented lacks coherence or purpose. The authors start with RNAseq analysis (without looking at eQTLs), then they integrate with human GWAS and human eQTL data, then they perform the DO GWAS, and after the DO eQTL. Finally, they focus on experimentally validating one locus of their choice. The manuscript may read much better if the flow is optimized. As a suggestion, one could first perform all mouse analyses, integrate with human data, and then finalize with the experimental validation.

As described in the introduction, there are two main limitations of human BMD GWASs; 1) causal gene discovery and 2) near exclusive focus on BMD. We split the manuscript

into two conceptual sections each of which address one of these limitations. The first section used mouse bone RNA-seq and a network-based approach to identify putatively causal genes. The second used GWAS in the DO to unravel the genetics of clinically relevant bone traits other than BMD. We believe this is the most logical presentation of the data. If we separated the sections, we feel we would lose the coherence of the two different goals of the work. However, in order to provide clarity, we have added a number of clarifying statements that provide more details as to exactly what analyses were performed and rationale for their inclusion.

- To make the manuscript more readable, it would be great to write down why a given analysis was performed. This is particularly important for those who have less computational background.

We have provided more detail regarding rationale for why specific assays were performed in the text.

- In general, it feels like most of the results are in supplementary figures or tables while the main figures and results convey very little information. This is especially the case for the RNA-seq part at the beginning, but also true for the DO QTL and eQTL data. On the same line, the supplementary tables are very difficult to understand and a graphic representation of the results will significantly help the reader to understand the data.

We believe the current figures and tables contain the most pertinent data. In the case of the “RNA-seq part in the beginning”, all of the most relevant data regarding our network analysis/colocalizing eQTL approach is provided in Table 1/Figure 3. In addition, Table 2 provides a list of all QTL identified in the DO along with important information (position, CI, number of missense variants, and genes with colocalizing eQTL). Most of the information in the supplement represents large datasets. It is not clear to us how these could be presented in a main text figure or table.

As for the difficulty in understanding the supplementary tables, we have now edited the tables and the table legends for clarity. We have also added two supplementary figures to provide further clarification of our computational approaches (Supplemental Figure 1 and Supplemental Figure 4).

- Are the 192 mice use for RNAseq also phenotyped with the other mice? Are they part of the 619 mice? They seem to be phenotyped based on supplementary table 8, but this should be clarified in the methods.

Yes, the 192 mice are a subset of the 619 mice, and were phenotyped with them. We have added a clarification of this to the first section of our Results, and to methods section “Bulk RNA isolation, sequencing and quantification”.

- The motivation behind the choice of computational strategy to identify BANs is not clear to me. Specifically, why did the authors choose to identify modules of co-expressed genes (using WGCNA signed co-expression network), followed by Bayesian Network construction on the genes of each module? There is little value (and it makes little sense) to perform Bayesian Network construction using features that are already positively correlated. Has this been done before?

Our analysis is based on previously published and widely used key driver analysis (KDA) approaches [19–22] (also cited within the main manuscript text). Because the analysis leverages the number of discrete connections between genes, up to a particular neighborhood size, Bayesian networks are excellent for our purpose, because as we learn the structure of a BN, we learn these connections between genes. WGCNA networks allow us to look at the overall gene-gene correlation structure at a high level, while BNs are sparser but allow a more granular look at the relationships between genes. This is because, unlike WGCNA, Bayesian networks are probabilistic “causal” networks, where the relationships between nodes imply a probabilistic belief that a node in the network is affecting other nodes. This logic is what drives our BAN analysis. In BN’s, connections between genes imply a causal relationship. That is to say, in a bone expression dataset, we can use BNs to identify genes that are more connected to “known bone genes” than we would expect by chance. Because of the implication of a causal relationship, being more connected to “known bone genes” than expected by chance implies that such a gene is more likely to be causally linked to other bone genes, and is therefore a putative bone gene in its own right.

Furthermore, the WGCNA analysis has two different purposes:

1- It allows us to cluster genes into categories (modules) that are linked by co-expression, and are therefore likely to be co-regulated and have similar functional effects. We can then correlate entire modules with traits, by summarizing all expression of genes within the module to an eigengene.

2-Learning Bayesian networks is very computationally intensive. By learning a Bayesian network for each cluster, we are able to more efficiently model relationships between genes. And since the genes in a module are previously correlated by co-expression (WGCNA), we are essentially learning the relationships between co-expressed genes in a more granular way than is possible with WGCNA alone, in which all genes are connected with all other genes, but in a weighted manner.

- Based on the author assumption, Qsox1 deletion should have resulted in a strong bone phenotype and this doesn’t seem to be the case. Can this come from the fact that the phenotyping data obtained from different CRISPR/Cas9 lines were combined? Or is it due to the mixed genetic background used (B6SJF1xC57BL/6J – random selection of 50 F2 male mice)?

We refer the reviewer to Figure 6 of our manuscript. As predicted from the association and eQTL analysis in the DO, *Qsox1* deletion resulted in a significant increase in ML overall ($P=1.8 \times 10^{-9}$) and in a sex-specific manner ($P=5.6 \times 10^{-7}$ and 3.5×10^{-3} for males and females, respectively). Furthermore, Figures 6H-I show significant increases in cortical bone morphology traits in *Qsox1* knockouts, namely the polar moment of inertia, maximum moment of inertia, and cortical area to total area ratio ($P=0.02$, 8.9×10^{-3} and 0.03 , respectively). Finally, Figure 6N of our manuscript shows that *Qsox1* deficiency significantly ($P=1.0 \times 10^{-3}$) increased femoral strength (max load).

Minor points:

- It may help that the computational strategies (which are in many cases custom) be explained graphically.

We agree. We have added two supplementary figures to graphically explain the two main custom computational strategies, which are the network analysis approach and the merge analysis. These two figures are now Supplemental Figure 1 and Supplemental Figure 4.

- Stylistic note: it is best to avoid nested parentheses (e.g. lines 66,68), as well as whole sentences in parentheses (e.g. line 67).

We agree. However, the aforementioned nested parentheses and sentence in parentheses are due to first-use of acronyms in the text.

- Line 155: The relevance of the following phrase is not clear to me: “indicating the structure of the Bayesian networks was not random with respect to connectivity”. Specifically, why would one expect that genes affecting bone traits to be more connected?

BNs model dependencies between nodes. Since these BNs are learned from clusters of co-expressed genes, and we are using expression data to learn BNs, we expect that there will be groups of more highly interconnected nodes in a BN, as those are groups of nodes that most likely strongly co-expressed. That is to say, their expression values are modeled as dependencies between nodes. This means that, in theory, you can begin to predict expression values for genes within these connections in a manner that is independent from non-descendant nodes. Therefore, in this case, genes with more connections can be thought of as genes with more “causal” links to other genes, based on expression values. Since the expression data in this case is from bone, we would expect that genes with more connections, which are more likely to be “involved” with many other genes (with which they are co-expressed), would be more likely to be bone-relevant genes. Basically, because we have a bone expression dataset, we expect bone genes to be more likely to affect the expression of other genes in that dataset than expected by chance, especially when constrained by co-expression, leading to more connections in the Bayesian networks. The relevance of the phrase is then confirmatory, as we are observing what we expect.

We have added a clarification of our rationale in the results (pg.8 lines 212-221)

- Line 161: was there correction for multiple testing?

No. We have clarified this by adding “nominal” to the text (pg. 8, line 231). The reason for not performing multiple testing corrections is mentioned in the discussion (pg. 25, lines 687-692). While a standard key driver analysis (KDA) approach would perform such a correction, we did not use the p-values to identify “key driver” genes. Instead, we used our BAN analysis to rank genes based on the likelihood that they are involved in a process, and chose to trim our list based on a nominal p-value, which we then followed up with eQTL-GWAS colocalization.

- Line 626: For the hypergeometric test, you refer to “total number of bone genes in our bone gene set”. Are these all expressed in your RNAseq dataset? It may be a good idea to remove those that are not expressed.

After carefully considering your point, we strongly agree. A hypergeometric test measures the significance of successful “draws” from the population containing a set number of “successes”, which in this case are “known bone genes”. In this case, hypergeometric tests were used to calculate the significance of the connectivity of a node with known bone genes, thereby highlighting that node as a BAN. Because the network data were generated from our RNA-seq dataset, it makes sense to only consider “known bone genes” that appear in that dataset. Otherwise, by using a list of “known bone genes” that includes non-expressed genes (the known bone list contains known regulators not expressed in bone per se), we are artificially inflating the number of “successes” within the population, as it is impossible for those non-expressed genes to be “drawn” (since they are not in the expression data, and therefore not in the networks). To rectify this, we trimmed our “known bone list” to only include genes expressed in our data (N=1,291). Because of this, we now have a greater number of BANs, and therefore a greater number of genes that colocalize. We have rectified the Methods and the BAN/BMD GWAS section to conform with these changes. While our previous approach was more conservative, we believe that this “fix” is more statistically “correct”.

- Line 238: There is no mention of how the correlations of phenotypes (in Supplementary Table 3) could affect the QTL mapping results. This is especially relevant for the Chromosome 1 locus that the authors define as “pleiotropic” (Line 294). Is it really pleiotropic because the locus is affecting multiple traits? Or because the traits are highly correlated?

This is a great point. We agree with you that it is not likely appropriate to use the term pleiotropic; though, it's not always straightforward and there will certainly be loci that have truly pleiotropic effects. As a result, we have modified the text (pg. 20-21, lines 519-526) to mention how correlations among traits are reflected in the GWAS.

- The selection of all GTEx tissues instead of a subset is not well explained. In addition, how do the data from different tissues overlap with human BMD GWAS? In other words, are there tissues that overlap more than others?

Great question. Because GTEx does not include bone tissues or cells, we chose to use all tissues. The rationale for this is the observation of sharing of eQTLs across tissues [5,6]. Therefore, a colocalizing eQTL in a non-bone tissue may represent a non-bone autonomous causal effect or it may simply reflect the actions of an eQTL active in bone and shared across non-bone tissues. It should also be noted that this is one advantage of the network-based approach we utilized, it relied on both network information and a colocalizing eQTL to implicate a gene as a candidate. We have clarified this rationale in the text (pg. 9, lines 279-284).

To address your second question, we identified all tissues where there was colocalization between GTEx eQTL and BMD GWAS. We observed significant colocalizations in 40 tissues. Some tissues have more colocalizing genes than others, such as tibial nerve and tibial artery, subcutaneous adipose, thyroid, lung and sun exposed skin. In general, of the tissues with significant colocalizing eQTL, brain tissues had the least colocalizations. We believe this shows the utility of using all tissues instead of a subset, especially when combined with other evidence like being a BAN and being in proximity of GWAS locus. We have added a supplementary file (S9) with all significant colocalizations and their tissues.

- You may want to better differentiate the human eQTLs and mouse eQTLs in the text to avoid confusion.

Great suggestion. We have made sure that it is clear for all instances of “eQTL” that it is clear whether we are talking about mouse or human eQTL.

- You should explain what PPH is, and why it is used.

We have added an explanation to the Methods under the “GWAS-eQTL colocalization” section, and have added a reference to that section within the Results.

CITATIONS:

1. Morris JA, Kemp JP, Youlten SE, Laurent L, Logan JG, Chai RC, et al. An atlas of genetic influences on osteoporosis in humans and mice. Nat Genet. 2019;51: 258–266. doi:10.1038/s41588-018-0302-x
2. Estrada K, Styrkarsdottir U, Evangelou E, Hsu Y-H, Duncan EL, Ntzani EE, et al. Genome-wide meta-analysis identifies 56 bone mineral density loci and reveals 14 loci associated with risk of fracture. Nat Genet. 2012;44: 491–501. doi:10.1038/ng.2249

3. Kurbatova N, Mason JC, Morgan H, Meehan TF, Karp NA. PhenStat: A Tool Kit for Standardized Analysis of High Throughput Phenotypic Data. *PLoS One*. 2015;10: e0131274. doi:10.1371/journal.pone.0131274
4. Shorter JR, Huang W, Beak JY, Hua K, Gatti DM, de Villena FP-M, et al. Quantitative trait mapping in Diversity Outbred mice identifies two genomic regions associated with heart size. *Mamm Genome*. 2018;29: 80–89. doi:10.1007/s00335-017-9730-7
5. GTEx Consortium. The GTEx Consortium atlas of genetic regulatory effects across human tissues. *Science*. 2020;369: 1318–1330. doi:10.1126/science.aaz1776
6. GTEx Consortium, Laboratory, Data Analysis & Coordinating Center (LDACC)—Analysis Working Group, Statistical Methods groups—Analysis Working Group, Enhancing GTEx (eGTEx) groups, NIH Common Fund, NIH/NCI, et al. Genetic effects on gene expression across human tissues. *Nature*. 2017;550: 204–213. doi:10.1038/nature24277
7. Friedman MA, Abood A, Senwar B, Zhang Y, Maroni CR, Ferguson VL, et al. Genetic Variability affects the Skeletal Response to Unloading. *Cold Spring Harbor Laboratory*. 2020. p. 2020.06.26.174326. doi:10.1101/2020.06.26.174326
8. Hisatomi Y, Kugino K. Changes in bone density and bone quality caused by single fasting for 96 hours in rats. *PeerJ*. 2019;6: e6161. doi:10.7717/peerj.6161
9. Devlin MJ. Why does starvation make bones fat? *Am J Hum Biol*. 2011;23: 577–585. doi:10.1002/ajhb.21202
10. Bathija A, Davis S, Trubowitz S. Bone marrow adipose tissue: response to acute starvation. *Am J Hematol*. 1979;6: 191–198. doi:10.1002/ajh.2830060303
11. Beamer WG, Donahue LR, Rosen CJ, Baylink DJ. Genetic variability in adult bone density among inbred strains of mice. *Bone*. 1996;18: 397–403. Available: <https://www.ncbi.nlm.nih.gov/pubmed/8739896>
12. Yuan JT, Gatti DM, Philip VM, Kasperek S, Kreuzman AM, Mansky B, et al. Genome-wide association for testis weight in the diversity outbred mouse population. *Mamm Genome*. 2018;29: 310–324. doi:10.1007/s00335-018-9745-8
13. Porcu E, Rüeger S, Lepik K, eQTLGen Consortium, BIOS Consortium, Santoni FA, et al. Mendelian randomization integrating GWAS and eQTL data reveals genetic determinants of complex and clinical traits. *Nat Commun*. 2019;10: 3300. doi:10.1038/s41467-019-10936-0
14. Brodie A, Azaria JR, Ofran Y. How far from the SNP may the causative genes be? *Nucleic Acids Res*. 2016;44: 6046–6054. doi:10.1093/nar/gkw500
15. Barbeira AN, Bonazzola R, Gamazon ER, Liang Y, Park Y, Kim-Hellmuth S, et al. Exploiting the GTEx resources to decipher the mechanisms at GWAS loci. doi:10.1101/814350
16. Lilue J, Doran AG, Fiddes IT, Abrudan M, Armstrong J, Bennett R, et al. Sixteen diverse laboratory mouse reference genomes define strain-specific haplotypes and novel functional loci. *Nat Genet*. 2018;50: 1574–1583. doi:10.1038/s41588-018-0223-8

17. Gatti DM, Svenson KL, Shabalin A, Wu L-Y, Valdar W, Simecek P, et al. Quantitative trait locus mapping methods for diversity outbred mice. *G3* . 2014;4: 1623–1633. doi:10.1534/g3.114.013748
18. Choi K, He H, Gatti DM, Philip VM, Raghupathy N, Gyuricza IG, et al. Genotype-free individual genome reconstruction of Multiparental Population Models by RNA sequencing data. Cold Spring Harbor Laboratory. 2020. p. 2020.10.11.335323. doi:10.1101/2020.10.11.335323
19. Watson CT, Cohain AT, Griffin RS, Chun Y, Grishin A, Hacyznska H, et al. Integrative transcriptomic analysis reveals key drivers of acute peanut allergic reactions. *Nat Commun*. 2017;8: 1943. doi:10.1038/s41467-017-02188-7
20. Huan T, Meng Q, Saleh MA, Norlander AE, Joehanes R, Zhu J, et al. Integrative network analysis reveals molecular mechanisms of blood pressure regulation. *Mol Syst Biol*. 2015;11: 799. doi:10.15252/msb.20145399
21. Mäkinen V-P, Civelek M, Meng Q, Zhang B, Zhu J, Levian C, et al. Integrative genomics reveals novel molecular pathways and gene networks for coronary artery disease. *PLoS Genet*. 2014;10: e1004502. doi:10.1371/journal.pgen.1004502
22. Wang I-M, Zhang B, Yang X, Zhu J, Stepaniants S, Zhang C, et al. Systems analysis of eleven rodent disease models reveals an inflammatome signature and key drivers. *Mol Syst Biol*. 2012;8: 594. doi:10.1038/msb.2012.24

Reviewers' Comments:

Reviewer #1:

Remarks to the Author:

Reviewer comments have been satisfactorily addressed

Reviewer #2:

Remarks to the Author:

The authors provided a very detailed rebuttal with a deep discussion of the criticisms raised. From my point of view, all aspects were sufficiently addressed either by providing additional information, corrections, or deletion respectively re-phrasing.

Reviewer #3:

Remarks to the Author:

I am satisfied that authors have responded to all of my initial concerns.

However, upon re-reading the manuscript, I believe more detail is required to ensure the reproducibility of the single cell data analysis. It is critical to clarify that these cells were differentiated *ex vivo* as this is currently not in the main text. I believe referring to these preparations as 'bone marrow stromal cells' may not be sufficient detail for the reader to grasp that these have been differentiated *ex vivo*. Further, if a standard osteoblastic differentiation protocol was used on the cells in culture, it should be stated as such in the methods as the term 'Bone marrow differentiation media' is somewhat confusing given the huge variety of cell lineages present in bone marrow. Moreover, the authors should give details as to the success of their procedure to generate mineralised osteoblasts, such as the extent of mineralisation, as this is important information to ascertain the maturity of the cells that are being sequenced and the composition of the . Analytically, details as to how cell types were defined (such as the specific gene markers used to define cell types) are critical to determine if the reader agrees with the authors cell type assignment and thus the localisation of the expression of genes of interest.

Further, I agree with the reviewer 4s statement that "For every dataset generated, the authors could perform some high level / conventional omics analysis." While the authors reply indicates this analysis is beyond the scope of the paper, I believe such analysis using established techniques would help orient the reader as to the quality of the data from a biological perspective (standard QC procedures typically focus on technical sources of noise). One of the key strengths of this paper are the resources it is contributing to the community (including the single cell dataset). More attention needs to be given to establishing the quality of these resources to maximise their utility.

Reviewer #4:

Remarks to the Author:

From the "bioinformatic perspective", the authors improved transparency and reporting by including references to raw and processed data, as well as code. They have also included supplementary figures depicting the computational strategies and explained the reasoning behind the different analyses.

Some remaining points:

- The new analysis pipeline diagrams are informative. I think that including them (albeit in a more compact form, with less whitespaces) within main figures could be a plus.
- It would still be very informative to show the PCA plots mentioned in the rebuttal, to visually show

the sources of variation.

- I still think that performing the differential analysis on the RNAseq would be great for prospective readers and the community.
- It is not clear how many of "the known genes" overlap with GWAS loci. Are known bone genes more likely to overlap with GWAS loci than a random set of genes?
- There are multiple and different references to the "known bone list" genes. For example: they are called "bona fide regulators of BMD" (line 696), 51 out of 72 genes being known regulators of bone traits (line 288). The problem is that the "known bone list" was generated from GO databases and then manually curated. It is not clear which GO domain(s) was/were used, although it seems it is "Biological Process". In addition, it is not clear what evidence codes were used, if any. For instance, some GO annotations are inferred from expression patterns (IEP). See this <http://geneontology.org/docs/guide-go-evidence-codes/>. Were evidence codes taken into account when constructing this list? The authors either have to refine some of the claims about these genes' implications in bone-related processes, modify the known bone list, or support the claims of these genes' direct involvement in bone processes.
- Minor: Figure 2 quality can be improved by reducing whitespaces, probably by laying the three panels horizontally in one row.
- Minor: line 692, "homed in" instead of "honed in"

From the "biological perspective", the authors did not make an effort in improving the biological relevance of their manuscript and did not follow my suggestions relative to this aspect. In particular:

- They still refer to the cells used in their study as BM-MSCs, even if they confirmed that they subjected the BM-MSCs to an osteoblast differentiation protocol. I agree that their culture is heterogeneous but they followed the protocol commonly used in the literature to obtain osteoblasts and this should be clearly indicated in the study. The authors should have used undifferentiated BM-MSCs to claim their conclusion.
- In addition, the scRNA-seq on this not clear cell population was performed on 4 females and only one male mice. This is clearly not balanced and might have introduced additional confounding factors to the study and to the interpretation of the results.
- They simply affirm the significance of the bone phenotype upon Qsox1 deletion, without making any effort in replying to my questions, which may enhance the effect of the deletion on the phenotype and improve the manuscript.

Response to Reviewers' critiques of Al-Barghouthi et al. "Systems genetic analysis in Diversity Outbred mice informs human bone mineral density GWAS and identifies Qsox1 as a novel determinant of bone strength"

We appreciate the additional comments from two of the reviewers. They raised excellent points and, as a result, the revised version of our manuscript is substantially improved. Below we address each of the critiques. Our responses are highlighted in bold red type following each critique. Additionally, all changes made in the manuscript have been tracked.

REVIEWER COMMENTS

Reviewer #3 (Remarks to the Author):

I am satisfied that authors have responded to all of my initial concerns.

However, upon re-reading the manuscript, I believe more detail is required to ensure the reproducibility of the single cell data analysis. It is critical to clarify that these cells were differentiated *ex vivo* as this is currently not in the main text. I believe referring to these preparations as 'bone marrow stromal cells' may not be sufficient detail for the reader to grasp that these have been differentiated *ex vivo*. Further, if a standard osteoblastic differentiation protocol was used on the cells in culture, it should be stated as such in the methods as the term 'Bone marrow differentiation media' is somewhat confusing given the huge variety of cell lineages present in bone marrow. Moreover, the authors should give details as to the success of their procedure to generate mineralised osteoblasts, such as the extent of mineralisation, as this is important information to ascertain the maturity of the cells that are being sequenced and the composition of the . Analytically, details as to how cell types were defined (such as the specific gene markers used to define cell types) are critical to determine if the reader agrees with the authors cell type assignment and thus the localisation of the expression of genes of interest.

Thank you for your comments. We changed all mentions of "bone marrow stromal cells" to "bone marrow stromal cells exposed to osteogenic differentiation media *in vitro*". We also clarified in the methods that the cells were subjected to a "standard *in vitro* osteoblast differentiation protocol", and changed "bone marrow differentiation media" to "osteogenic differentiation media" (pg. 52, lines 1012-1013).

As for the details of our success in generating mineralizing osteoblasts, we have included a plot of the mineralization of differentiated cells across time from each of the 5

samples (referenced in main text on pg. 16, lines 298-299, Supplemental Table 13 and Supplemental Figure 2), and have added details to the methods (Pg. 52, lines 1015-1025).

In order to aid the readers in how cell types were defined, we included the results of differential expression analysis for each of the clusters, which we used to manually annotate the cell-type of each cluster. These data are now included as Supplemental Table 14, and we have added details to the methods (pg. 54, lines 1062-1065).

Further, I agree with the reviewer 4s statement that "For every dataset generated, the authors could perform some high level / conventional omics analysis." While the authors reply indicates this analysis is beyond the scope of the paper, I believe such analysis using established techniques would help orient the reader as to the quality of the data from a biological perspective (standard QC procedures typically focus on technical sources of noise). One of the key strengths of this paper are the resources it is contributing to the community (including the single cell dataset). More attention needs to be given to establishing the quality of these resources to maximise their utility.

Thank you for this comment. For the scRNA-seq data, we performed all high-level/conventional -omics analyses that are based on established techniques/protocols. While we only show results that are relevant to the manuscript in the main figures, all data, computational scripts and results of these analysis are available as supplementary tables or as raw data. As mentioned in the previous comment above, we have now also included the results of the differential expression analysis on scRNA-seq data (Supplemental Table 14), which we used to annotate cell-type identity for each cluster.

After receiving this round of reviews, it seems that the only high-level/omics analysis that we were missing were a differential expression and PCA analyses on the cortical bone RNA-seq data. As we document in the response to reviewer #4's comments below, we have now performed these analyses (pg 7-8, lines 160-170) and have provided the results as Supplemental Figure 1 and Supplemental Tables 5 and 6. Details have been added to the methods (pg 38, ln 698-712).

Reviewer #4 (Remarks to the Author):

From the "bioinformatic perspective", the authors improved transparency and reporting by including references to raw and processed data, as well as code. They have also included supplementary figures depicting the computational strategies and explained the reasoning behind the different analyses. Some remaining points:

- The new analysis pipeline diagrams are informative. I think that including them (albeit in a more compact form, with less whitespaces) within main figures could be a plus.

We agree. We have now included them as main Figures 3 and 5.

- It would still be very informative to show the PCA plots mentioned in the rebuttal, to visually show the sources of variation.

Thank you. The PCA plots are now presented as Supplemental Figure 1, and are referenced in the main text (pg. 7, lines 160-164). We have also added details on how the PCA was performed in the methods (pg. 38, lines 699-701).

- I still think that performing the differential analysis on the RNAseq would be great for prospective readers and the community.

Thank you. We performed differential expression analyses based on sex and bone strength (max load) (by comparing high and low bone strength) and have referenced the results in the main text (pg. 7-8, lines 164-170). The results of these analyses are now presented in Supplemental Tables 5 and 6. We have also added details on how the analyses were performed to the methods (pg. 39, lines 701-712)

- It is not clear how many of “the known genes” overlap with GWAS loci. Are known bone genes more likely to overlap with GWAS loci than a random set of genes?

To answer this, we performed a Fisher’s Exact Test based on the number of “known bone genes” with human homologs (1,270) which overlapped with the GWAS loci +/- 1Mbp (771). Based on the number of protein-coding genes which also overlapped these loci (10,809), we find that GWAS loci are enriched in “known bone genes” (OR = 1.35, $P=1.45^{-7}$). Therefore, known bone genes are more likely to overlap with GWAS loci than a random set of genes.

We have added these data to the results (pg. 9, lines 209-211).

- There are multiple and different references to the “known bone list” genes. For example: they are called “bona fide regulators of BMD” (line 696), 51 out of 72 genes being known regulators of bone traits (line 288). The problem is that the “known bone list” was generated from GO databases and then manually curated. It is not clear which GO domain(s) was/were used, although it seems it is “Biological Process”. In addition, it is not clear what evidence codes were used, if any. For instance, some GO annotations are inferred from expression patterns (IEP). See this <http://geneontology.org/docs/guide-go-evidence-codes/>. Were evidence codes taken into account when constructing this list? The authors either have to refine some of the claims about these genes’ implications in bone-related processes, modify the known bone list, or support the claims of these genes’ direct involvement in bone processes.

Thank you. We have corrected our oversight in the methods, and have now clarified that we used all three GO domains, and did not consider evidence codes (pg. 41, lines 757-759). Furthermore, we’d like to emphasize that we also used the MGI Human-Mouse: Disease Connection database in constructing the “known bone gene” list. This allowed us to identify genes whose perturbation *in vivo* impacts bone. In fact, of the 1,291 genes,

only 444 of the genes are uniquely drawn from the GO data. Overall, we believe that generating such a list from these public resources is standard procedure.

We also agree that the claims we used to characterize these genes implications were strong, and have softened our language and clarified throughout the text that the genes in the “known bone gene” list are putative bone genes, and that the 51 BANs are “putative regulators of bone traits”.

- Minor: Figure 2 quality can be improved by reducing whitespaces, probably by laying the three panels horizontally in one row.

Thank you for this suggestion. We have rearranged Figure 2 to reduce whitespace.

- Minor: line 692, “homed in” instead of “honed in”

Thanks. We have changed this line to say “We then identified genes most likely...” (pg 29, line 519)

From the “biological perspective”, the authors did not make an effort in improving the biological relevance of their manuscript and did not follow my suggestions relative to this aspect. In particular:

- They still refer to the cells used in their study as BM-MSCs, even if they confirmed that they subjected the BM-MSCs to an osteoblast differentiation protocol. I agree that their culture is heterogeneous but they followed the protocol commonly used in the literature to obtain osteoblasts and this should be clearly indicated in the study. The authors should have used undifferentiated BM-MSCs to claim their conclusion.

Thank you. We changed all mentions of “bone marrow stromal cells” to “bone marrow stromal cells exposed to osteogenic differentiation media *in vitro*”. We have also clarified that we followed a standard *in vitro* osteoblast differentiation protocol (methods pg. 52, lines 1011-1012). Please refer to our response to Reviewer 3 above for additional details.

- In addition, the scRNA-seq on this not clear cell population was performed on 4 females and only one male mice. This is clearly not balanced and might have introduced additional confounding factors to the study and to the interpretation of the results.

We have now provided supplementary figures of the UMAP plots, split by sex (Supplemental Figure 3), and have referenced these plots in the main body of the results (pg. 16-17, lines 304 and 306). As can be seen, the expression of Sertad4 and Gltd2 localize to the same clusters in both sexes, indicating that, as far as the interpretation of our results, there doesn't seem to be any confounding due to sex.

- They simply affirm the significance of the bone phenotype upon *Qsox1* deletion, without making any effort in replying to my questions, which may enhance the effect of the deletion on the phenotype and improve the manuscript.

We apologize for not addressing your critique. This was not our intention. Here is your prior critique relating to the data on *Qsox1* knockout mice:

- Based on the author assumption, *Qsox1* deletion should have resulted in a strong bone phenotype and this doesn't seem to be the case. Can this come from the fact that the phenotyping data obtained from different CRISPR/Cas9 lines were combined? Or is it due to the mixed genetic background used (B6SJF1xC57BL/6J – random selection of 50 F2 male mice)?

I believe confusion may have stemmed from our thought that you were questioning whether the observation that *Qsox1* deletion affected bone morphology. However, it seems you were referring to a difference in the relative strength of the QTL in the DO compared to what was observed in *Qsox1* KO mice.

We agree that there is a difference between the effect sizes. To quantify this we focused on medial-lateral femoral (ML) width. First, we assumed the ML QTL was driven by a single biallelic QTL (with WSB/EiJ harboring the ML increasing allele and all other strains possessing the ML decreasing allele). To estimate the effect on ML of a single WSB/EiJ allele, we compared mean ML in WSB/EiJ homozygotes to the average of the mean values in heterozygotes with one WSB/EiJ allele (we didn't compare other homozygotes because the N per genotype was typically small). The observed difference due to replacement of one WSB/EiJ was -0.064 mm. Given the additive nature of the QTL, we would expect the effect of replacing two WSB/EiJ alleles to be -0.128 mm. As you can see in Figure 8C the overall differences between wild-type and *Qsox1* homozygous mutant mice is 0.064 mm (1.829 mm - 1.765 mm). So yes, you are correct the effect is weaker in *Qsox1* knockouts, especially when you consider a lack of QSOX1 activity in the KO mice versus the reduction of *Qsox1* expression (and likely activity) in the DO.

We believe there are multiple potential explanations for the observation. First, as you suggest it could involve genetic background. It is possible that there are alleles in the DO that interact with the Chr. 1 QTL. We think it is unlikely to be an effect of the SJLxB6 background per se (relative to a pure B6 background for example), though this cannot be ruled out. We also cannot rule out an effect of combining data from different mutants, though we also think this is unlikely given that they all abolish QSOX1 activity and we observed similar phenotypes in each individual line. Another explanation would be that the effects of the QTL and *Qsox1* knockout are different because they are not equivalent. For example, we have demonstrated that the locus is complex and likely harbors at least two QTL. It is quite possible that there are other variants in the region altering bone morphology, of which the *Qsox1* eQTL explains part of the aggregate effect.

We have added text to the discussion describing the observed difference (pg. 31, lines 558-568).

Reviewers' Comments:

Reviewer #3:

Remarks to the Author:

I am satisfied that the authors have addressed all comments and concerns and thank them for their contribution to the field.

Reviewer #4:

Remarks to the Author:

The authors now addressed all of our comments.

Response to Reviewers' critiques of Al-Barghouthi et al. "Systems genetic analysis in Diversity Outbred mice informs human bone mineral density GWAS and identifies Qsox1 as a novel determinant of bone strength"

We appreciate the informative and thoughtful comments from all four reviewers. They raised excellent points and, as a result, the revised version of our manuscript is substantially improved. Below we address each of the critiques. Our responses are highlighted in bold red type following each critique. Additionally, all changes made in the manuscript have been tracked.

REVIEWERS' COMMENTS

Reviewer #3 (Remarks to the Author):

I am satisfied that the authors have addressed all comments and concerns and thank them for their contribution to the field.

We thank the reviewer for their comments.

Reviewer #4 (Remarks to the Author):

The authors now addressed all of our comments.

We thank the reviewer for their comments.